# Agro-Industrial Wastewater Treatment with *Acacia dealbata* Coagulation/Flocculation and Photo-Fenton-Based Processes

Nuno Jorge [1,*], Ana R. Teixeira [2], Marco S. Lucas [2] and José A. Peres [2]

1   Escuela Internacional de Doctorado (EIDO), Campus da Auga, Campus Universitário de Ourense, Universidade de Vigo, As Lagoas, 32004 Ourense, Spain
2   Centro de Química de Vila Real (CQVR), Departamento de Química, Universidade de Trás-os-Montes e Alto Douro (UTAD), Quinta de Prados, 5000-801 Vila Real, Portugal
*   Correspondence: njorge@uvigo.es

**Abstract:** The removal of dissolved organic carbon (DOC) and total polyphenols (TPh) from agro-industrial wastewater was evaluated via the application of coagulation–flocculation–decantation (CFD) and Fenton-based processes. For the CFD process, an organic coagulant based on Acacia dealbata Link. leaf powder (LP) was applied. The results showed that the application of the LP at pH 3.0, with an LP:DOC ratio of 0.5:1 (*w/w*), achieved a high removal of turbidity, total suspended solids (TSS), and volatile suspended solids (VSS) of 84.7, 79.1, and 76.6%, respectively. The CFD sludge was recycled as fertilizer in plant culture (germination index ≥ 80%). Afterwards, the direct application of Fenton-based processes to raw WW was assessed. The Fenton-based processes (UV/Fenton, UV/Fenton-like, and heterogeneous UV/Fenton) showed high energy efficiency and a cost of 1.29, 1.31 and 1.82 €/g/L DOC removal, respectively. The combination of both processes showed the near complete removal of TPh and DOC after 240 min of reaction time, with high energy efficiency. In accordance with the results obtained, the combination of CFD with Fenton-based processes achieves the legal limits for the disposal of water into the environment, thus allowing the water to be recycled for irrigation.

**Keywords:** fenton-based processes; leaves powder; sludge recycling; winery wastewater; water recycling

## 1. Introduction

The agricultural and industrial sectors are responsible for 4 and 28%, respectively, of the world's gross domestic product. The main agro-industrial products are derived from sugarcane, cassava, cereals, oilseeds, grapes, milk, derivates, and animal slaughter [1]. Among these sectors, the wine industry demarcates itself as one of the largest producers worldwide, with a global income of EUR 25.8–31.4 billion between 2014 and 2018, and a production volume of between $2.49 \times 10^{10}$ and $2.92 \times 10^{10}$ L [2]. To produce a quality wine, it is necessary to wash and disinfect all the equipment from the winery, such as the presses used for crushing the grapes, fermentation tanks, and barrels, among other equipment components [3]. This sanitation step produces large volumes of winery wastewater, rich in soluble sugars, ethanol, glycerol, esters, organic acids, phenols, and various populations of bacteria and yeast, with an acidic pH (between 3–4) [4]. If the WW was to be released into the environment without treatment, it can cause the pollution of water, degradation of the soil, damage to the vegetation, release odor and air emissions, and cause the eutrophication of water resources [5]. In this work, we propose the treatment of these types of winery wastewaters (WWs) by different methodologies: (1) coagulation–flocculation–decantation (CFD), (2) Fenton-based processes, and (3) CFD combined with Fenton-based processes (Figure 1).

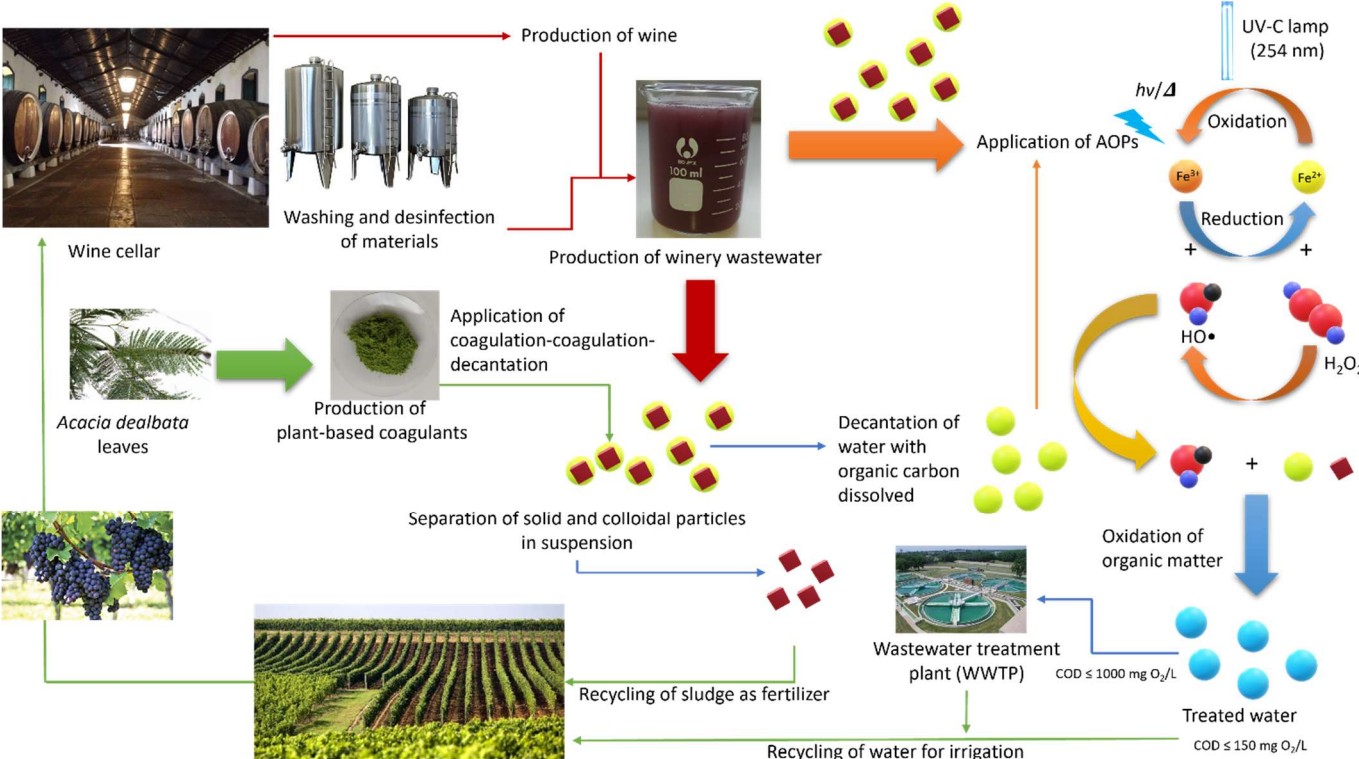

**Figure 1.** Winery valorization cycle: wine production, wastewater treatment coagulation–flocculation–decantation, advanced oxidation processes (AOPs), and the recycling of sludge and treated water.

The CFD process is mostly used to achieve solid–liquid separation, in which the small solids and colloidal suspended particles are destabilized by the addition of coagulants/flocculants, are agglomerated in flocs, and finally, precipitation occurs. The CFD process has the advantage of its cost-effectiveness and ease of operation [6,7]. In the literature, many authors attempt to apply new ecological coagulants, such as plant-based coagulants, for the treatment of wastewater. For example, in the work of Heredia and Martín [8], a tannin-based flocculant was used to remove heavy metals from polluted surface water. In the work of Jorge et al. [9], plant extracts were applied to decrease the costs of microfiltration, and in the work of Renault et al. [10], a chitosan flocculant was applied to treat cardboard-mill wastewater. In the work of Martins et al. [11], several plant-based coagulants were used to treat agro-industrial wastewater. In this work, we propose the production of plant-based coagulants derived from *Acacia dealbata* Link. leaves. *A. dealbata* was first introduced in Europe in 1816 for horticultural and floricultural purposes (in Portugal, 1850), although in 1999 it was officially listed as invasive [12]. Several authors point out the successful use of coagulants derived from different subspecies of the plant *Acacia* to treat wastewater [13–17]; therefore, *A. dealbata* leaves could provide a good product for CFD processes, although, to our knowledge, no studies on this have been reported. Therefore, two environmental problems are addressed in this work: the treatment and control of WW via *A. dealbata* dissemination. *A. dealbata* leaves can be successively transformed into plant-based coagulants to treat WW and the sludge could be recycled as fertilizer since, in this case, no toxic materials are being used. Treated water can also be recycled for irrigation, thus, two problems become one solution.

The dissolved organic carbon present in WW can be treated with the application of advanced oxidation processes (AOPs). AOPs are processes based on the generation of hydroxyl radicals (HO$^\bullet$) with an oxidation potential of 2.8 V that can react with a pollutant and oxidize it into the simpler intermediates, $CO_2$ and $H_2O$ [18,19]. Among AOPs, the Fenton process can be applied, which consists of the catalytic decomposition of hydrogen peroxide ($H_2O_2$) by ferrous iron ($Fe^{2+}$) to generate HO$^\bullet$ radicals [20]. However, some

authors use ferric iron ($Fe^{3+}$) instead, and the reaction is called a Fenton-like process. This is a slow process since the reaction of $Fe^{3+}$ with $H_2O_2$ generates hydroperoxyl radicals ($HO_2^{\bullet}$), in accordance with Equation (1), which are much less reactive due to a lower oxidation potential (1.70 V) [21,22].

$$Fe^{3+} + H_2O_2 \rightarrow Fe^{2+}HO_2^{\bullet} + H^+ \quad\quad (1)$$

The Fenton and Fenton-like processes have some drawbacks, such as the need to recover the iron after the treatment and the requirement to operate within a limited pH range (3–4); therefore, heterogeneous catalysts can be applied to overcome these problems [23]. Ferrocene was applied with success in the treatment of wastewater contaminated with methylene blue [24,25]. However, ferrocene was never applied in the treatment of WW, and this complex matrix cannot be compared to the simple matrix of a wastewater contaminated with methylene blue. To increase the efficiency of Fenton-based processes, the application of UV radiation promotes the regeneration of $Fe^{3+}$ to $Fe^{2+}$ and increases the kinetic rate of organic carbon degradation. The photo-Fenton process is effective for the degradation of emerging contaminants [26,27], textile dyes [28,29], insecticides [30] etc.; however, it has been pointed out in the work of Rodríguez-Chueca et al. [31] that the matrix of the WW is very complex, making this type of wastewater more difficult to treat. This difficulty lies with the presence of alcohols (mainly ethanol), $Cl^-$, $HCO_3^-$, and $CO_3^{2-}$ ions, which can act as radical scavengers [32].

The main aim and novelty of the present work were to develop a plant-based coagulant from *A. dealbata* leaves, separate the solid parts (sludge), and recycle them as fertilizer. This work also aims to evaluate the efficiency of several Fenton-based processes with and without radiation for the reduction of organic carbon and the reuse of the water for irrigation. Finally, this work purposes to evaluate the effect of the different treatments regarding kinetic rate, energy efficiency, and operational cost.

## 2. Materials and Methods

### 2.1. Reagents and Winery Wastewater Sampling

Ferrous sulfate heptahydrate ($FeSO_4 \bullet 7H_2O$) was acquired from Panreac, Barcelona, Spain; ferrocene (($C_5H_5)_2Fe$) was acquired by Alfa Aesar, Ward Hill, MA, USA; iron(III) chloride hexahydrate ($FeCl_3 \bullet 6H_2O$) was acquired by Merck, Darmstadt, Germany. Hydrogen peroxide ($H_2O_2$ 30% *w/w*) and Titanium(IV) oxysulfate solution, 1.9–2.1%, for the determination of hydrogen peroxide were acquired by Sigma-Aldrich, St. Louis, MI, USA. Sodium sulfite anhydrous ($Na_2SO_3$) was acquired from Labkem, Barcelona, Spain. For pH adjustment, sodium hydroxide (NaOH) from Labkem, Barcelona, Spain and sulphuric acid ($H_2SO_4$, 95%) from Scharlau, Barcelona, Spain, were used. Deionized water was used to prepare the respective solutions. The WW was collected from a cellar located in the Douro region (northern Portugal). The wastewater samples were placed in plastic containers to be transported to the laboratory and stored at −40 °C until further use.

### 2.2. Analytical Determinations

Different physical-chemical parameters were determined to characterize the WW, including turbidity, total suspended solids (TSS), volatile suspended solids (VSS), chemical oxygen demand (COD), biological oxygen demand ($BOD_5$), dissolved organic carbon (DOC), and total polyphenols (TPh). The main WW characteristics are shown in Table 1.

**Table 1.** Principal characteristics of the agro-industrial wastewater.

| Parameters | WW |
|---|---|
| pH | $4.0 \pm 0.2$ |
| Electrical conductivity (µS/cm) | $62.5 \pm 3.1$ |
| Turbidity (NTU) | $296 \pm 5.9$ |
| Total suspended solids—TSS (mg/L) | $750 \pm 15$ |
| Volatile suspended solids—VSS (mg/L) | $640 \pm 12.8$ |
| Chemical Oxygen Demand—COD (mg $O_2$/L) | $2145 \pm 21.5$ |
| Biochemical Oxygen Demand—$BOD_5$ (mg $O_2$/L) | $550 \pm 5.5$ |
| Dissolved Organic Carbon—DOC (mg C/L) | $400 \pm 8.0$ |
| Total Nitrogen—TN (mg N/L) | $9.07 \pm 0.5$ |
| Total polyphenols—TPh (mg gallic acid/L) | $22.6 \pm 1.1$ |
| Biodegradability index—$BOD_5$/COD | $0.26 \pm 0.03$ |
| Aluminum (mg/L) | $0.01 \pm 0.001$ |
| Calcium (mg/L) | $1.07 \pm 0.1$ |
| Cobalt (mg/L) | $0.01 \pm 0.001$ |
| Copper (mg/L) | $0.014 \pm 0.001$ |
| Iron (mg/L) | $0.05 \pm 0.001$ |
| Magnesium (mg/L) | $0.51 \pm 0.01$ |
| Manganese (mg/L) | $0.016 \pm 0.002$ |
| Potassium (mg/L) | $20.5 \pm 1.0$ |
| Sodium (mg/L) | $0.19 \pm 0.04$ |
| Zinc (mg/L) | $10.53 \pm 0.2$ |

Total polyphenols were measured by the Folin-Ciocalteau method, adapted by Singleton and Rossi [33]; chemical oxygen demand (COD) analysis was carried out in a COD reactor from Macherey-Nagel (Düren, Germany) and a HACH DR 2400 spectrophotometer (Loveland, CO, USA) was used for colorimetric measurements. Biochemical oxygen demand ($BOD_5$) was determined using a respirometric OxiTop® IS 12 system (WTW, Yellow Springs, OH, USA), total nitrogen (TN) and DOC samples were analyzed by direct injection of the filtered samples into a Shimadzu TOC-L$_{CSH}$ analyzer (Shimadzu, Kyoto, Japan), equipped with an ASI-L autosampler, provided with an NDIR detector and calibrated with standard solutions of potassium phthalate. Hydrogen peroxide concentration was followed using titanium (IV) oxysulfate (DIN 38 402H15 method) at 410 nm, using a portable spectrophotometer from Hach (Loveland, CO, USA). The iron concentrations were analyzed by atomic absorption spectroscopy (AAS) using a Thermo Scientific™ iCE™ 3000 Series (Thermo Fisher Scientific, Waltham, MA, USA). Turbidity was determined by a 2100N IS Turbidimeter (Hach, Loveland, CO, USA). The total suspended solids were determined by a portable spectrophotometer (Hach, Loveland, CO, USA). VSSs were determined according to Standard Method 2540E, using a glass fiber membrane (VWR European, retention of 1.6 µm). pH was determined by a 3510 pH meter (Jenway, Cole-Parmer, UK) and conductivity was determined by a portable condutivimeter, VWR C030 (VWR, V. Nova de Gaia, Portugal), in accordance to the methodology of the Standard Methods [34].

*2.3. Leaves Powder Preparation*

The leaves used in this work were collected from *Acacia dealbata* Link., in University of Trás-os-Montes and Alto Douro (UTAD), Vila Real, Portugal, at GPS location 41°17′12.9″ N 7°44′14.1″ W, during May 2022, and transported to the Environmental Engineering Laboratory of UTAD, where they were stored until used. The leaves were washed and dried in an oven at 70 °C for 24 h. Then, they were grounded into powder using a groundnut miller. The grounded powder was then sieved to a mesh size of 150 µm. Finally, the powder was once more dried in an oven at 70 °C for 30 min to remove the moisture. The powder was left to cool and stored in a tightly closed plastic jar [16]. The leaves powder presents a pH in water of $6.21 \pm 0.2$ (Figure 2).

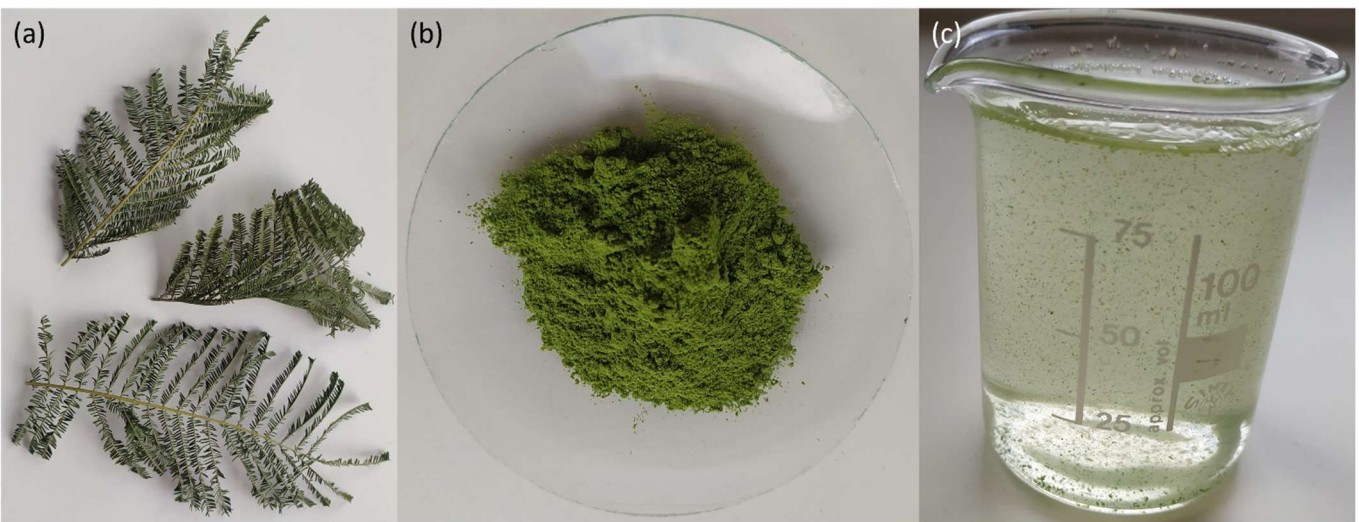

**Figure 2.** (**a**) *A. dealbata* leaves, (**b**) leaves powder (LP), and (**c**) LP in water.

### 2.4. Leaves Powder Characterization

The chemical spectra profile of the LP was obtained by Fourier-transform infrared spectroscopy (FTIR), with the mixing of 2 mg plant powder with 200 mg KBr. The powder mixture was inserted into molds and pressed at 10 ton/cm$^2$ to obtain a transparent pellet. The sample was analyzed with an IRAffinity-1S Fourier-transform infrared spectrometer (Shimadzu, Kyoto, Japan) and the infrared spectra in transmission mode were recorded in the 4000–400 cm$^{-1}$ frequency region. The microstructural characterization of LP was carried out with a scanning electron microscopy (FEI QUANTA 400 SEM/ESEM, Fei Quanta, Hillsboro, WA, USA) and the chemical composition of the powder was estimated using energy dispersive X-ray spectroscopy (EDS/EDAX, PAN'alytical X'Pert PRO, Davis, CA, USA). The specific surface area (S$_{BET}$) of LP was determined by applying the Gurevitsch's rule at a relative pressure p/p$_0$ = 0.30 and according to the Brunauer, Emmet, Teller (BET) method from the linear part of the nitrogen adsorption isotherms. Different pore volumes were determined by the Barrett, Joyner, and Halenda model (BJH model).

### 2.5. Coagulation-Flocculation-Decantation Experimental Setup

The CFD process was performed in 800 mL beakers, which were placed in a Jar-Test device (ISCO JF-4, Louisville, KY, USA), with four mechanic agitators, powered by a regulated speed engine (Figure 3). The mixture of the LP with WW samples was performed under a fast mix of 150 rpm/ 3 min, and slow mix of 20 rpm/20 min, at an ambient temperature (298 K). The samples were taken at periodic intervals (15 min, 1, 2, 4, 6, 8, and 12 h) and the CFD process was optimized as follows:

(1) The pH was varied (3.0, 5.0, 7.0, 9.0, and 11.0) under the following conditions: DOC = 400 mg C/L, [LP]:DOC = 1:1 (*w/w*);

(2) The ratio LP:DOC was varied (0.25:1, 0.5:1, 1:1, 2:1, and 5:1 *w/w*) under the following conditions: pH = 3.0, DOC = 400 mg C/L.

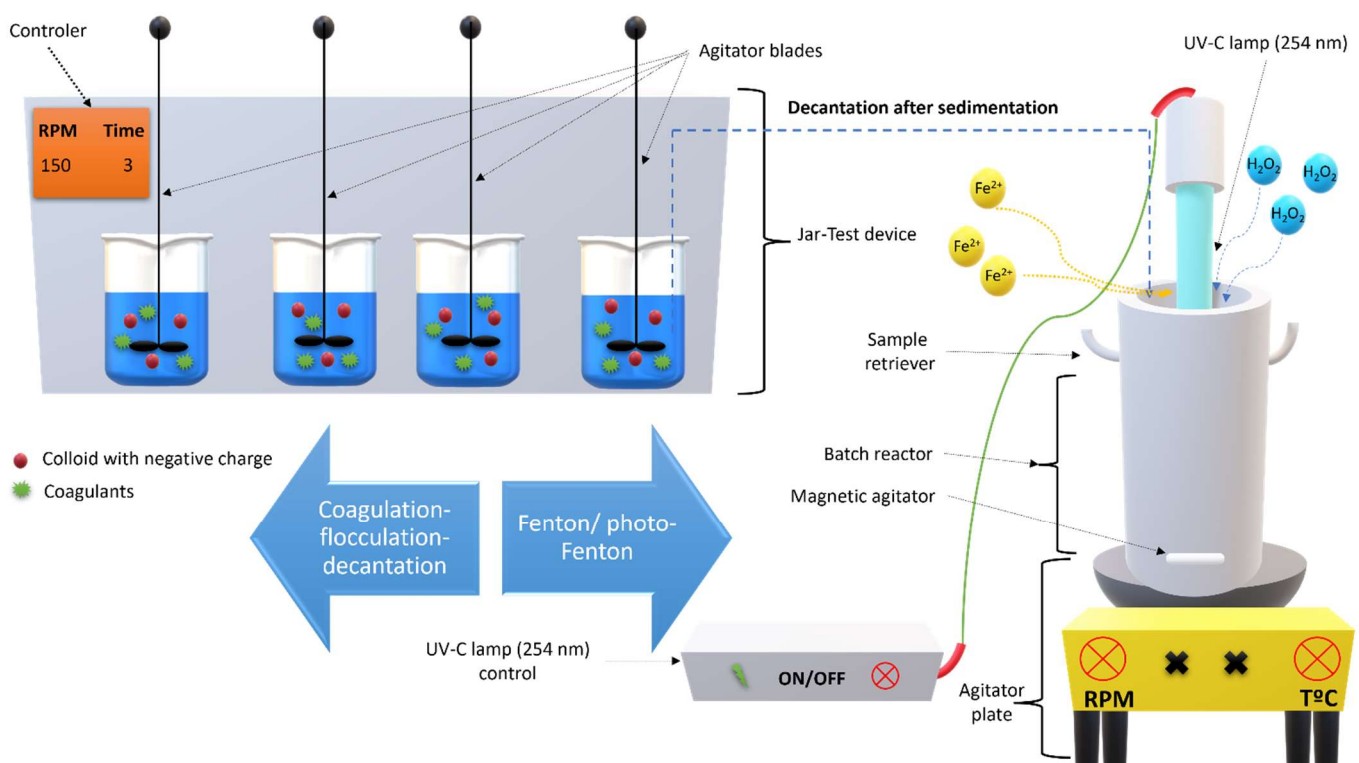

**Figure 3.** Experimental setup of CFD process and oxidation reactor.

### 2.6. Fenton Based Experiments Setup

The Fenton-based experiments were conducted in a reactor (height 18 cm, diameter 8 cm), with a volume of 500 mL. The cylinder reactor is made of borosilicate glass with ports in the upper section for sampling (Figure 3). As a radiation source, a *Heraeus* (Hanau, Germany) TNN 15/32 lamp (14.5 cm in length and 2.5 cm in diameter) mounted in the axial position inside the reactor was employed, with a spectral output of 253.7 nm (85–90%) and 184.9 nm (7–10%). The Fenton-based experiments were conducted under the operational conditions as follows: pH = 3.0, $[Fe^{2+}] = [Fe^{3+}] = 2.5$ mM, temperature = 298 K, agitation = 350 rpm, time = 240 min. The pH and the iron concentrations were selected based on studies performed in winery wastewater treatment, in which the pH varied from 3.0 to 9.0, and the $Fe^{2+}$ concentration varied from 0.5 to 10 mM (data not shown). To perform heterogeneous Fenton and heterogeneous photo-Fenton processes, a ferrocene catalyst was used because ferrocene is a $Fe^{2+}$ inducer that does not dissolve in wastewater. At the end of the experiments, sodium sulfite anhydrous was applied to quench the reactions and the pH was raised to 7.0 to remove the iron from the wastewater.

To determine the removal percentage of the parameters, Equation (2) was applied as follows [35,36]:

$$X_i(\%) = \frac{C_i - C_F}{C_i} \times 100 \qquad (2)$$

where, $C_i$ and $C_f$ are the initial concentrations, and 100 is the conversion factor.

### 2.7. Phytotoxicity Test

Phytotoxicity tests were performed via the germination of the *Raphanus sativus* (radish) and *Cucumis sativa* (cucumber) seeds (standard species recommended by the US Environmental Protection Agency, the US Food and Drug Administration, and the Organization for Economic Cooperation and Development [37]). Seeds were immersed in a 10% sodium hypochloride solution for 10 min to ensure surface sterility, then they were soaked in pure water. One piece of filter paper (Whatman filter paper 9 cm, Maidstone, UK) was put into

each 100 mm × 15 mm Petri dish, and 5 mL of test medium was added [38]. Seeds were then transferred onto the filter, with 10 seeds per dish and a 1 cm or larger distance between each seed. Petri dishes were covered and sealed with tape and placed in a controlled atmosphere with a constant temperature (25 °C), maintained during the course of the experiment by a WTM TS 608-G/2-i (Weilheim, Germany). After 7 days of darkness and 7 with light, the germination index was determined by Equation (3), in accordance with Varnero et al. [39] and Tiquia and Tam [40], as follows:

$$\text{GI}(\%) = \frac{\overline{N}_{SG,T}}{\overline{N}_{SG,B}} \times \frac{\overline{L}_{R,T}}{\overline{L}_{R,B}} * 100 \tag{3}$$

where GI is the germination index, $\overline{N}_{SG,T}$ is the arithmetic mean of the number of germinated seeds in each extract (wastewater), $\overline{N}_{SG,B}$ is the arithmetic mean of the number of germinated seeds in standard solution (distilled water), $\overline{L}_{R,T}$ is the mean root length of each extract (wastewater), and $\overline{L}_{R,B}$ is the mean root length in control (distilled water). If GI ≤ 50%, then there was a high concentration of phytotoxic substances, if 80% < GI > 50%, then there was a moderate presence of phytotoxic substances, and if GI ≥ 80%, then there were no phytotoxic substances (or they exist in very small dosages).

### 2.8. Statistical Analysis

Statistical analysis was performed using a one-way analysis of variance (ANOVA) and differences were considered as significant when $p < 0.05$, with average values compared using Tukey's test. The statistical analyses were performed using OriginLab 2019 software (Northampton, MA, USA). All data are presented as mean and standard deviation (mean ± SD).

## 3. Results and Discussions

From Table 1, it can be observed that the winery wastewater presents a low biodegradability index ($BOD_5$/COD = 0.26); thus, a chemical strategy treatment could be more suitable. The first part of this work consists in the study of performances of the CFD processes, employing a novel plant-based coagulant (LP), which shows great capacity to reduce the turbidity and TSS of WW. Considering that this work deals only with vegetal material, the sludge proved to be reusable for fertilizing. Then, different Fenton-based processes were applied directly to the raw WW, and results showed that photo-Fenton, photo-Fenton-like, and heterogeneous photo-Fenton processes could reach COD values that allowed the WW to be disposed as domestic wastewater to a WWTP. The combination of CFD with Fenton-based processes reached the Portuguese legal values for wastewater discharge into the environment; thus, the treated water can be recycled for irrigation. In the following sections this work is explained in more detail.

### 3.1. Leaves Powder Characterization

Figure 4 shows the Fourier-transform infrared spectroscopy (FTIR) spectrum of the LP, from which the area between 4000 and 400 cm$^{-1}$ was analyzed. The peaks observed were similar to the peaks observed in tannins from *Acacia dealbata* bark. It is observed that there is a band at 3423.65 cm$^{-1}$, indicating the presence of phenolic hydroxyl groups, proteins, fatty acids, carbohydrates, and lignin (OH stretching vibration). The bands at 2920.22 and 2848.86 cm$^{-1}$ indicate the presence of fatty acids (CH and $CH_2$ vibrations). The bands at 1658.78, 1516.05, and 152.39 cm$^{-1}$ are attributed to aromatic ring stretching vibrations. The band at 1035.77 cm$^{-1}$ is attributed to C-O stretching vibration [41].

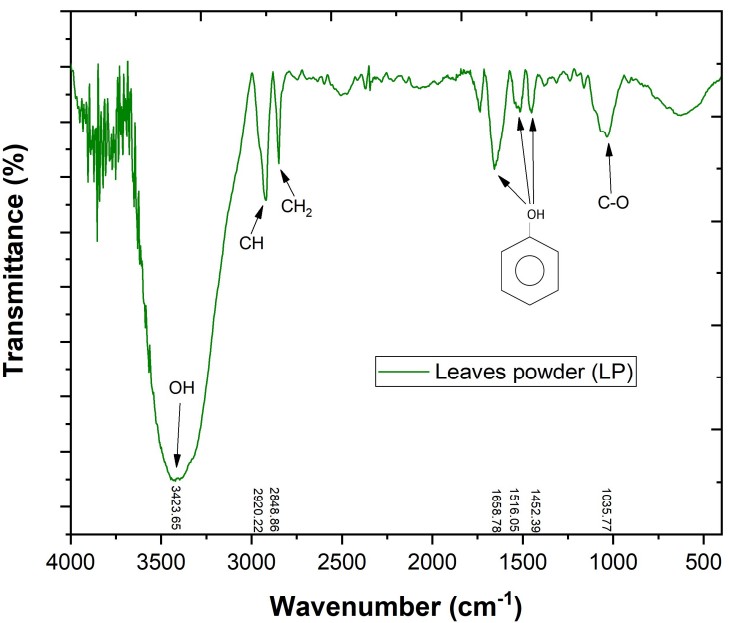

**Figure 4.** Fourier-transform infrared spectroscopy (FTIR) analysis of LP.

Figure 5a shows a scanning electron microscope (SEM) image of LP, using an ampliation of 250×. The dark spaces indicate the existence of pores, which is important because they facilitate the process of adsorption and they provide a high internal surface area, which could contribute to the decrease in turbidity and total suspended solids in the WW [42]. An EDS analysis (Figure 5b) revealed that carbon and oxygen were the main atoms in the LP composition, with small amounts of magnesium, aluminum, phosphorous, sodium, chloride, potassium, and calcium, which are used by the plant for stress resistance and photosynthesis [43]. The combination of both analyses suggests that LP is a carbon-based material with an adsorption capacity.

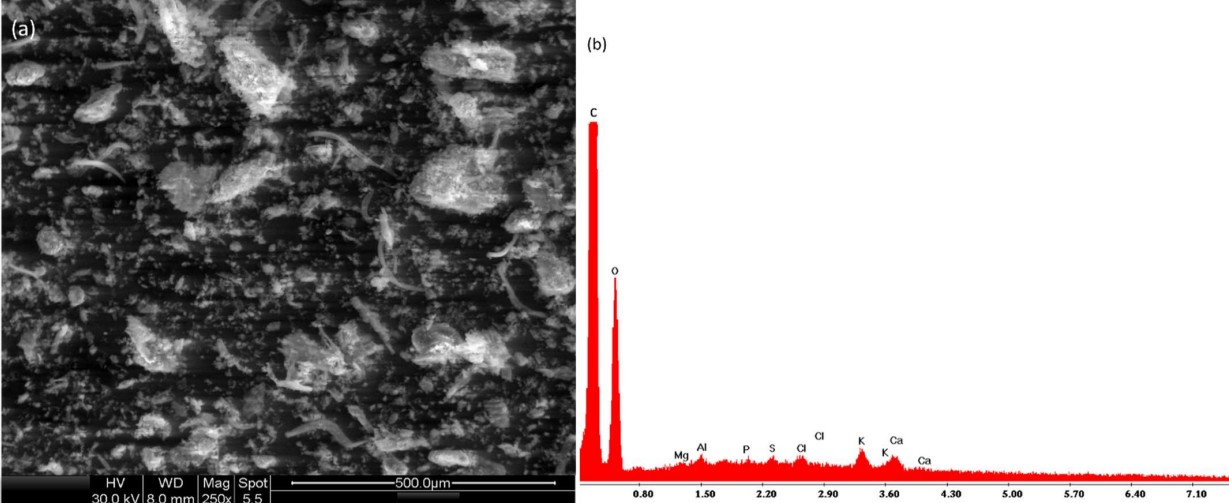

**Figure 5.** (**a**) Scanning electron microscopy (SEM) image of LP; (**b**) energy dispersive spectroscopy (EDS) analysis of LP.

The BET analysis showed that the nitrogen adsorption–desorption isotherms of LP are of type I isotherms, typical of microporous solids having relatively small external surfaces as defined by the International Union of Pure and Applied Chemistry (IUPAC) classification [44]. It also shows a $S_{BET}$ of 0.08 m$^2$/g, which is typical of these types of materials without mesoporosity.

### 3.2. Coagulation–Flocculation–Decantation Experiments

#### 3.2.1. Effect of pH Variation

Considering the low pH of the WW (between 3–4), it is necessary to determine how the pH influences the efficiency of the *A. dealbata* leaves coagulant; therefore, the pH of the WW was varied, using a range between 3.0 and 11.0. The results showed the highest decrease of DOC occurred under the application of pH 3.0 (11.7%), and this decreased as the pH increased (Figure 6a). The TPh was analyzed, and the results show no considerable removal after 12 h of sedimentation time (Figure 6b). The VSS analysis indicates the evolution of biomass removal via LP (Figure 6c). The results showed the highest removal occurred with an application of pH 3.0 (75.0%), with this decreasing under increasing pH. These results are consistent with the results obtained for turbidity and TSS (Figure 6d), which showed the highest removal occurred with the application of pH 3.0 (78.3 and 74.7%, respectively). The COD removal was also analyzed, reaching the highest removal percentage at pH 3.0 (12.4%). As previously observed by EDS and FTIR analysis, water-soluble cations are available in the LP coagulant in the form of proteins, and the results obtained by VSS removal indicate that the LP has a similar action mechanism to *Moringa Oleifera* seeds, in which the proteins work as natural polyelectrolytes. The mechanism used by LP involves adsorption and charge neutralization due to the porous structure (as observed in the SEM image), adsorption and interparticle bridging, and precipitation and enmeshment [45,46].

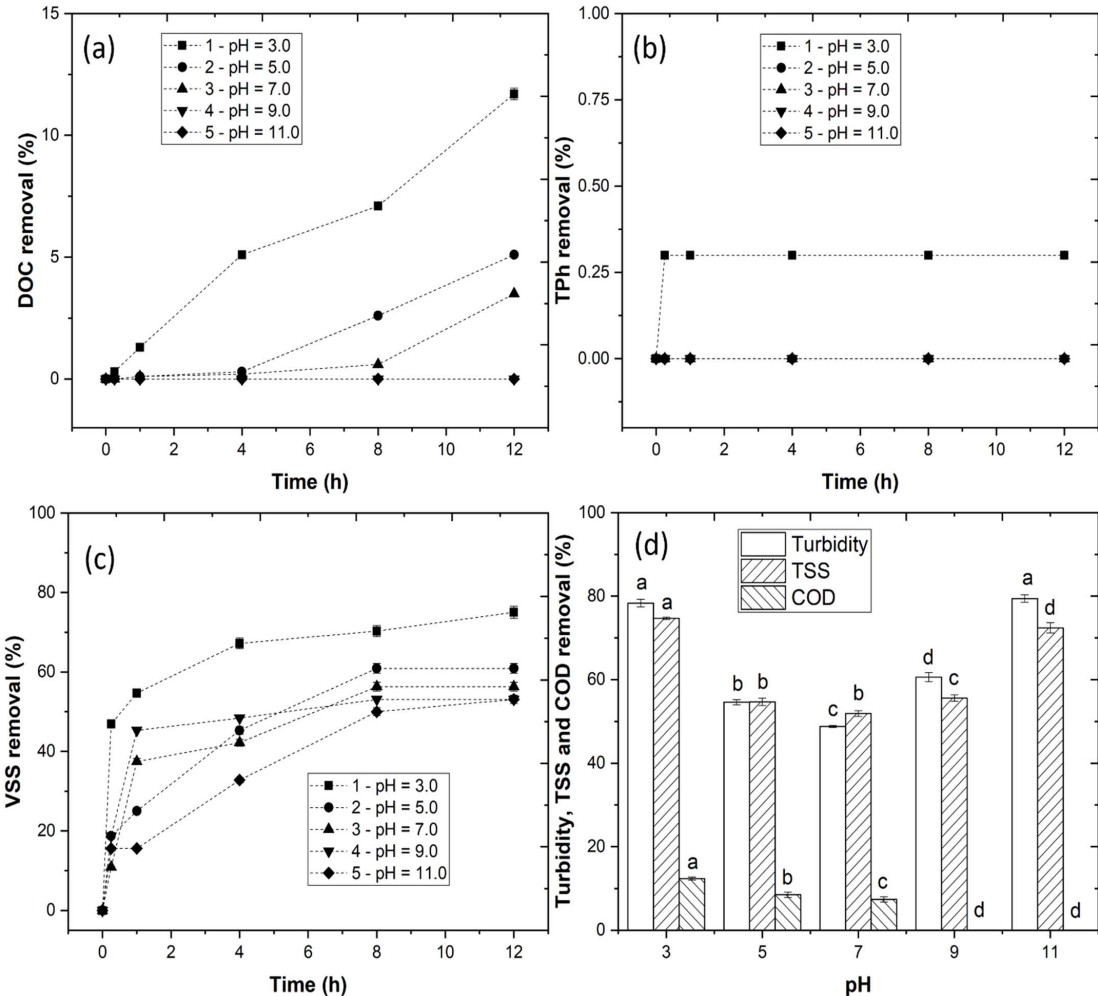

**Figure 6.** Effect of pH on (**a**) DOC removal, (**b**) TPh removal, (**c**) VSS removal, and (**d**) turbidity, TSS, and COD removal. CFD operational conditions were as follows: DOC = 400 mg C/L, LP:DOC = 1:1 *w/w*, V = 500 mL, temperature 298 K, fast mix 150 rpm/3 min, slow mix 20 rpm/20 min, and sedimentation time 12 h. Means in bars with different letters represent significant differences ($p < 0.05$) within each analysis by comparing wastewater pH.

The results obtained under the application of pH 5.0, 7.0, and 9.0 could be related to the deprotonation of the proteins that exist in the LP, leading to electrostatic repulsion and resulting in a decrease in turbidity, TSS, and VSS removal. These results were similar to those of Agbovi and Wilson [47], who observed a deprotonation of amphoteric chitosan flocculants under alkaline pH, resulting in lower turbidity removal.

3.2.2. Effect of LP Concentration

In the previous section, the high removal of turbidity, TSS, and VSS was observed; however, the removal of the other parameters remained low. This low removal could be associated with the concentration of the LP used. In the work of Chung et al. [48], it was observed that a variation in peanut–okra and wheat germ–okra concentration influenced the removal of the turbidity, TSS, and COD of palm oil mill effluent. In this section, the LP concentration effect was studied, changing the LP:DOC ratio ($w/w$) from 0.25:1 to 5:1, maintaining the initial pH of 3.0. The results showed an increase in DOC removal when the LP:DOC ratio varied from 1:1 to 0.5:1 (18.2%). For LP:DOC ratios higher than 0.5:1, DOC removal decreased (Figure 7a). These results could be due to the desorption of organic material from the LP to the wastewater, which slows the removal of dissolved organic matter. TPh analysis was performed (Figure 7b), considering the presence of polyphenols in the leaves. The results showed that the application of LP:DOC ratios of 0.25:1 and 0.5:1 achieved the highest TPh removal, with 67.5 and 35.8%, respectively. Above this ratio, the removals were not significant. In order to prove this idea, different concentrations of LP were mixed in water, and the DOC and TPh concentrations were measured over time. The results show that the LP releases and absorbs organic carbon during the 12 h (Figure 7e). It is also shown that after 12 h, an LP concentration of 200 mg/L has the lowest DOC concentration in water, thus explaining the highest DOC removal observed in the CFD process with a 0.5:1 LP:DOC ratio. In Figure 7f, TPh concentration in water is shown. The results show that increasing the LP concentration leads to a significant increase in the TPh concentration in water. These results explain the low removal of TPh observed with the application of LP:DOC ratios higher than 0.5:1.

The biomass removal results showed a high removal under the application of a 0.5:1 LP:DOC ratio (76.6%). The biomass results were consistent with those of the turbidity and TSS removal, which were observed to be significant (Figure 7c); thus, the LP acted mainly on the biodegradable fraction suspended in the WW. These results were in agreement with Rizzo et al. [49], who observed that the application of higher dosages of coagulant lead to the removal of turbidity and TSS from WW. The removal of this biodegradable fraction influenced the COD content. As observed in Figure 7d, an LP:DOC ratio of 0.5:1 showed the most significant COD removal, with 21.0%. The application of higher ratios showed lower COD removals, which was expected considering the low DOC, TPh, and VSS removal observed.

3.2.3. Water and Sludge Recycling

A resulting product of the CFD process is sludge due to flocs sedimentation. In accordance with the pH or the concentration of the coagulant, more compacted sludge can be produced, leading to a recovery of more of the supernatant water. Figure 8 shows the volume of sludge produced and water recovery as a function of the WW pH and the LP:DOC ratio. The results showed a significant compression of the sludge at pH 3.0 (41 mL/L), leading to a higher water recovery (91.8%). With the variation of the LP:DOC ratio, lower LP concentrations lead to higher sludge compression. With the application of a 5:1 LP:DOC ratio, the sludge compaction significantly decreased, possibly due to the sedimentation of higher amounts of the coagulant. These results differed from those obtained by authors such as Verna et al. [50], who used inorganic coagulants ($FeCl_3$) to treat petrochemical wastewater and observed that higher concentrations were required to obtain a more compacted sludge. In the work of Amuda and Amoo [51], it was noted that high dosages of polyelectrolytes were necessary (for application with $FeCl_3$) to produce more compact sludges in the treatment of beverage industrial wastewater. This work shows that the application of plant-based coagulant LP achieves highly

compacted sludges without requiring the addition of high concentrations of the coagulant or polyelectrolytes.

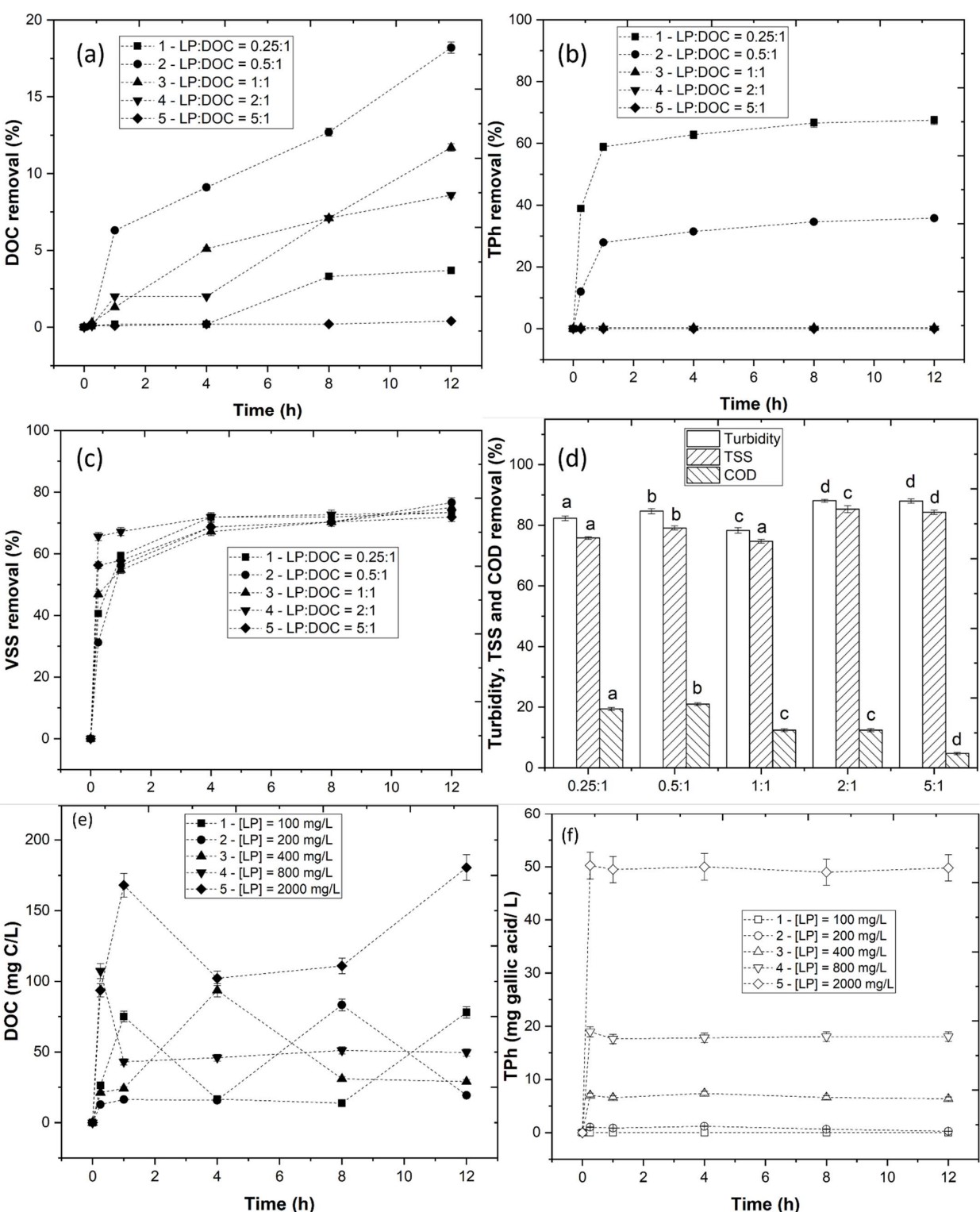

**Figure 7.** Effect of LP concentration on (**a**) DOC removal, (**b**) TPh removal, (**c**) VSS removal, and (**d**) turbidity, TSS, and COD removal, (**e**) DOC desorption in water, (**f**) TPh desorption in water. CFD operational conditions were as follows: pH = 3.0, DOC = 400 mg C/L, V = 500 mL, temperature 298 K, fast mix 150 rpm/3 min, slow mix 20 rpm/20 min, sedimentation time 12 h. Means in bars with different letters represent significant differences ($p < 0.05$) within each analysis by comparing LP:DOC ratios.

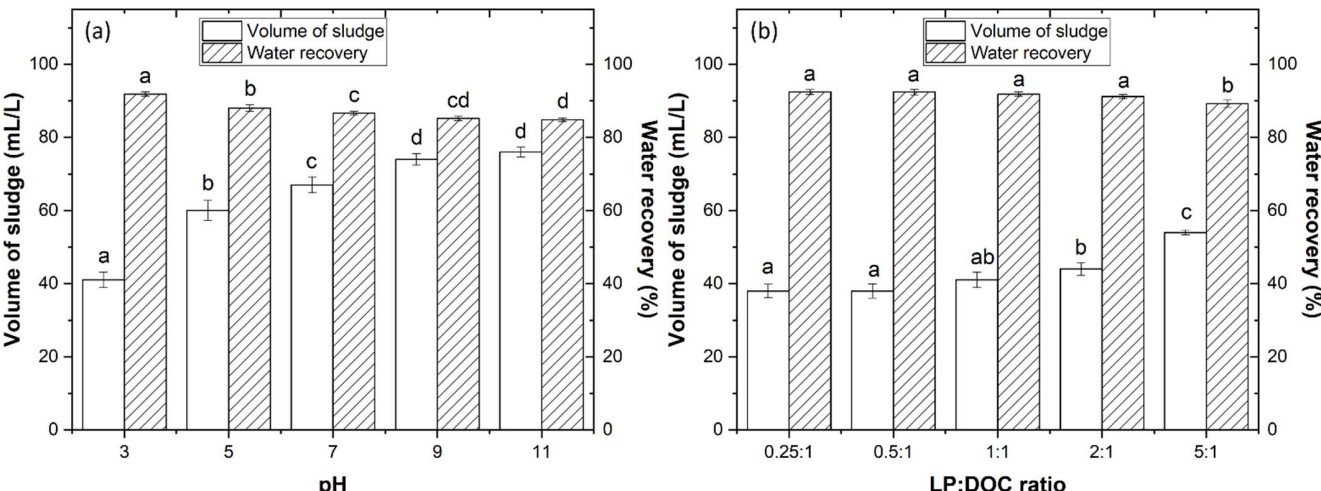

**Figure 8.** Assessment of the volume of sludge produced and the recovery of water under (**a**) different pH and (**b**) LP:DOC ratios. Means in bars with different letters represent significant differences ($p < 0.05$) within each parameter (volume of sludge and water recovery) by comparing wastewater pH and LP.DOC ratios.

The CFD process allowed the separation of the water from the solid part (sludge). Considering the mineral and organic composition of the sludge, it was proposed in this work that the WW sludge could be used as a fertilizer for agriculture. Therefore, after the CFD process, the sludge was collected and applied as a fertilizer for the germination of cucumber and radish seeds. The results in Figure 9a showed that the change in pH did not affect the root development of the seeds. It was also observed, for radish seeds, that the application of acidic pH achieved a high root length in comparison to an alkaline pH. The germination index (GI) results showed values >80% (Figure 9b); therefore, the final sludge showed low toxicity. The LP:DOC ratio was evaluated, and results showed longer root length with the application of a lower LP:DOC ratio (Figure 9c). These results suggested that the LP composition could, to some extent, inhibit the growth of the seeds. An evaluation of GI showed values > 80% (Figure 9d); therefore, the application of sludge as a fertilizer is deemed safe. Although, in the work of Ioannou et al. [5], it was described that the acidic pH of the WW can affect plant vigor by reducing the availability of plant nutrients and decreasing the populations of useful microbes. The results obtained in this work suggest that the WW sludge can be recycled as a fertilizer. These results are in agreement with Flores et al. [52], who observed low toxicity in WW after treatment in constructed wetlands.

### 3.3. Fenton-Based Experiments

#### 3.3.1. Effect of Iron Compounds and $H_2O_2$

In the previous section, it was observed that the application of LP reduced a limited amount of the DOC and TPh contents. Therefore, to decrease the remaining organic matter in the WW, it was necessary to find a complementary process. As a preliminary work, three different iron sources (ferrous sulfate, ferric chloride, and ferrocene) were added to the WW. $H_2O_2$ was also added by itself to study the same effect.

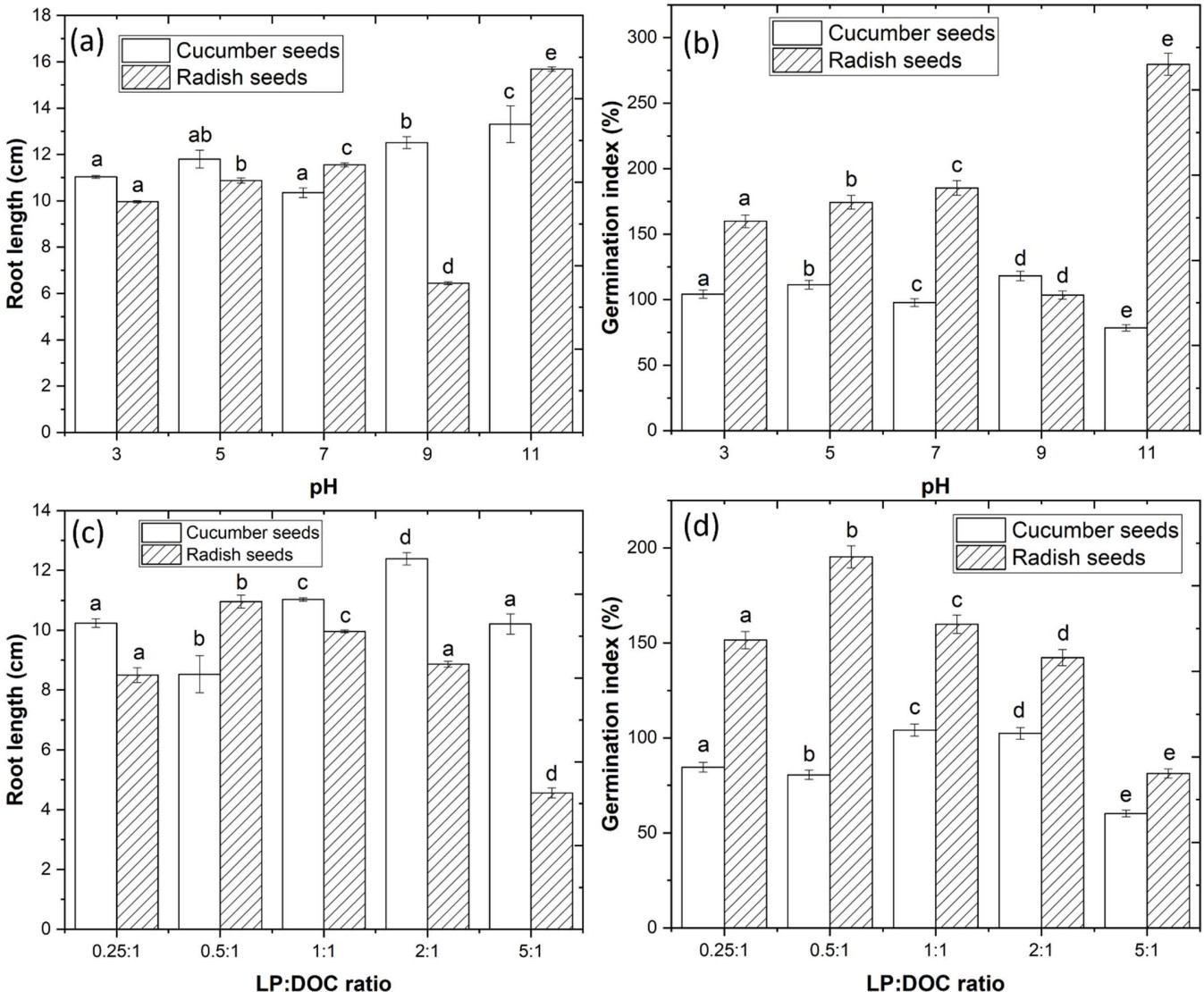

**Figure 9.** Effect of pH on (**a**) root length and (**b**) the germination index (GI) of cucumber and radish seeds. Effect of LP:DOC ratio on (**c**) root length and (**d**) the germination index (GI) of cucumber and radish seeds. Means in bars with different letters represent significant differences ($p < 0.05$) within each seed (cucumber and radish) by comparing wastewater pH and LP.DOC ratios against root length and germination index.

With the application of the three iron compounds, some DOC removal was observed, reaching a maximum of 19.0, 4.9 and 2.7%, respectively, for ferrous sulfate, ferric chloride, and ferrocene. The TPh were also analyzed, and the results showed the removal of 0.5<, 12.1 and 14.9%, respectively (Figure 10). The application of these compounds was previously observed in the works of Ilias et al. [53] and Ishak et al. [54] as coagulants. These compounds, after a fast and slow mixture, generate flocs with high dimensions, which then precipitate. However, in this work, there was constant agitation for 240 min, which led to the fragmentation of these flocs; thus, a lower DOC and TPh were observed. When ferrocene was applied, a physical process of adsorption was observed due to the porous nature of ferrocene. A similar mechanism was observed in the work of Guimarães et al. [55], in which the application of heterogeneous catalysts in the treatment of WW showed an 18% TOC removal after 240 min.

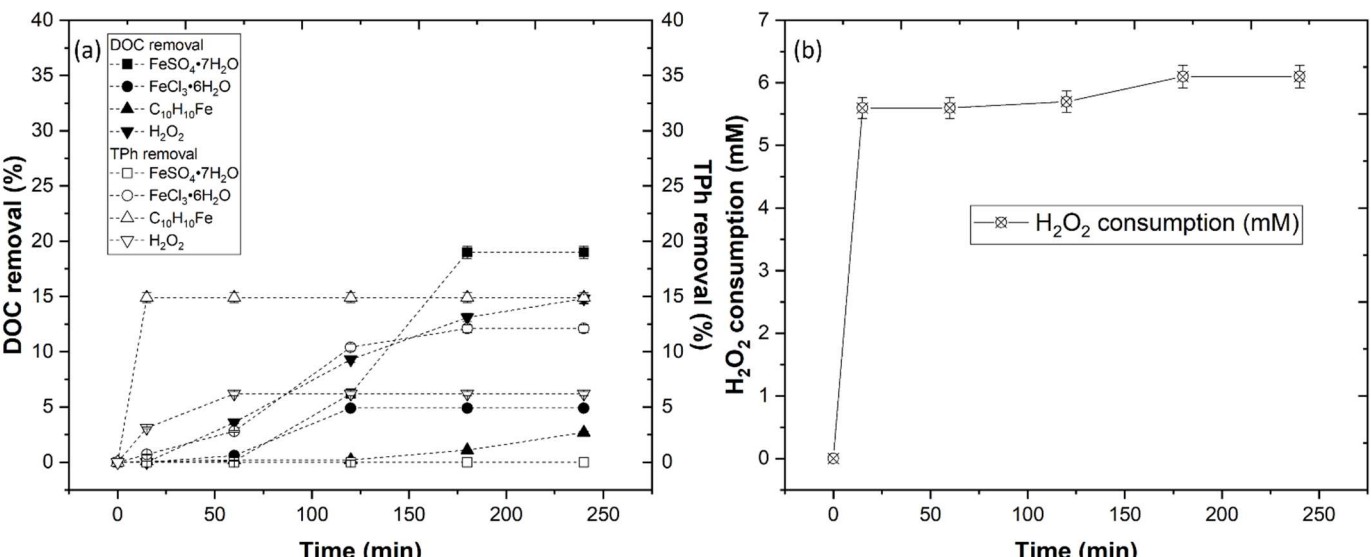

**Figure 10.** (**a**) Removal of DOC and TPh with the application of iron compounds $FeSO_4 \cdot 7H_2O$, $FeCl_3 \cdot 6H_2O$, $C_{10}H_{10}Fe$ and $H_2O_2$ alone. (**b**) $H_2O_2$ consumption. Operational conditions were as follows: pH = 3.0, $[Fe^{2+}] = [Fe^{3+}]$ = 2.5 mM, temperature = 298 K, agitation 350 rpm, time = 240 min.

$H_2O_2$ was also added (alone) to understand the efficiency of the oxidant in DOC and TPh removal. The results showed a removal percentage of 14.8 and 6.2%, respectively. In the work of Martins et al. [11], different AOPs were applied to degrade agro-industrial wastewater, and it was observed that $H_2O_2$ had a lower efficiency regarding the application of a catalyst alone, similar to the results obtained in this work; however, after 240 min, a DOC and TPh removal of 14.8 and 6.8%, respectively, was recorded. This removal could be attributed to the oxidation potential of $H_2O_2$ ($E^{\circ}$ = 2.77 V), which is able to degrade more sensible organic matter.

### 3.3.2. Fenton, Fenton-like, and Heterogeneous Fenton

Considering the low efficiency of the iron compounds and $H_2O_2$ alone, Fenton-based processes were studied to assess DOC and TPh reduction in WW. From Figure 11a, a fast decrease of DOC can be observed within the first 60 min, reaching a plateau at 240 min of 26.9, 30.7, and 12.2%, respectively, for Fenton, Fenton-like, and heterogeneous Fenton. Fenton and Fenton-like achieved a high removal of TPh (63.9 and 97.1%) in comparison to heterogeneous Fenton (<0.5%). These results are in agreement with Esteves et al. [21], who tested Fenton and Fenton-like processes in the treatment of olive mill wastewater (OMW) and observed a considerable removal of TPh. A DOC removal of 44 and 41% for Fenton and Fenton-like, respectively, was also observed, similar to the results obtained in this work. These results can be explained by the generation of $HO^{\bullet}$ radicals due to the reaction of $Fe^{2+}$ and $Fe^{3+}$ with $H_2O_2$ (Equation (4)) [56], leading to a higher degradation of organic matter. However, the introduction of ferrous sulfate or ferric chloride also led to the introduction of $SO_4^{2-}$ and $Cl^-$ ions, which are known to scavenger $HO^{\bullet}$ radicals, which decrease the reaction efficiency, as observed by Deng et al. [57]; thus, explaining the slow DOC removal after 60 min of reaction.

$$Fe^{2+} + H_2O_2 \rightarrow Fe^{3+} + HO^{\bullet} + HO^- \tag{4}$$

The heterogeneous Fenton showed lower DOC and TPh removal in comparison to the Fenton and Fenton-like processes. Ferrocene is a $Fe^{2+}$ inducer, contributing to the rapid generation of $HO^{\bullet}$ radicals; however, the catalyst released the iron into the solution in smaller amounts, thus a lower production of $HO^{\bullet}$ radicals was observed. These results differ from those in works such as Zhang et al. [58], who observed the degradation of 4-chlorophenol (4-CP) with the application of heterogeneous Fenton using ferrocene as a catalyst. The wastewater used by Zhang et al. [58] is made only by the contaminant dissolved in the water; however, these agro-industrial wastewater matrixes are very complex, making their treatment more difficult.

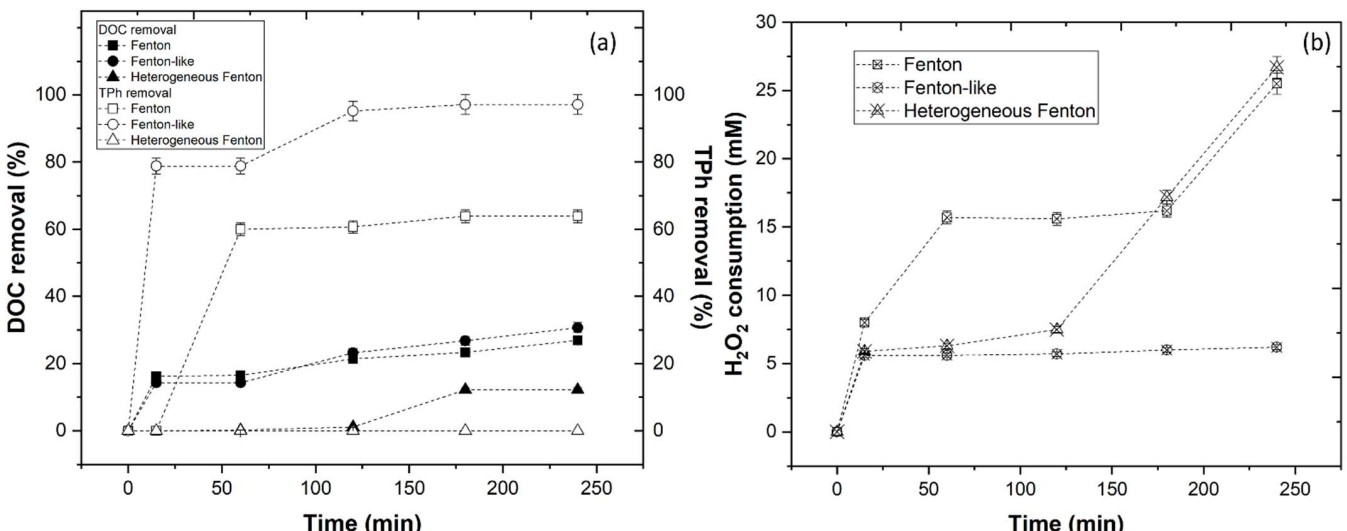

**Figure 11.** (**a**) Removal of DOC and TPh with the application of Fenton, Fenton-like, and heterogeneous Fenton. (**b**) $H_2O_2$ consumption. Fenton-based operational conditions were as follows: pH = 3.0, $[Fe^{2+}] = [Fe^{3+}] = 2.5$ mM, temperature = 298 K, agitation 350 rpm, time = 240 min.

### 3.3.3. Effect of UV-C and UV-C + $H_2O_2$

To improve the efficiency of the Fenton-based processes, UV radiation can be employed. In the literature, authors point out the application of UV-C, UV-A, and solar radiation can be used to complement Fenton-based processes. In this work, we studied the effect of UV-C radiation and UV-C/$H_2O_2$ for DOC and TPh removal from WW. Figure 12 shows that UV-C had a low effect on DOC removal, achieving 7.2% after 240 min. The application of UV-C + $H_2O_2$ showed a higher DOC removal of 17.4%. When TPh removal was evaluated, the results showed the near complete removal of it after 240 min with the application of UV-C/$H_2O_2$; however, only 20.7% was removed using UV-C alone. These results showed that polyphenols are more sensible to radiation; however, the remaining organic matter is practically not affected.

In comparison to Fenton, Fenton-like and heterogeneous Fenton, UV-C + $H_2O_2$ had a higher consumption of $H_2O_2$. DOC and TPh removal, achieved with the application of UV-C + $H_2O_2$, can be explained by the generation of $HO^{\bullet}$ radicals due to the conversion of $H_2O_2$ by the UV-C radiation (Equation (5)) [35].

$$H_2O_2 + h\nu \; \rightarrow \; 2HO^{\bullet} \tag{5}$$

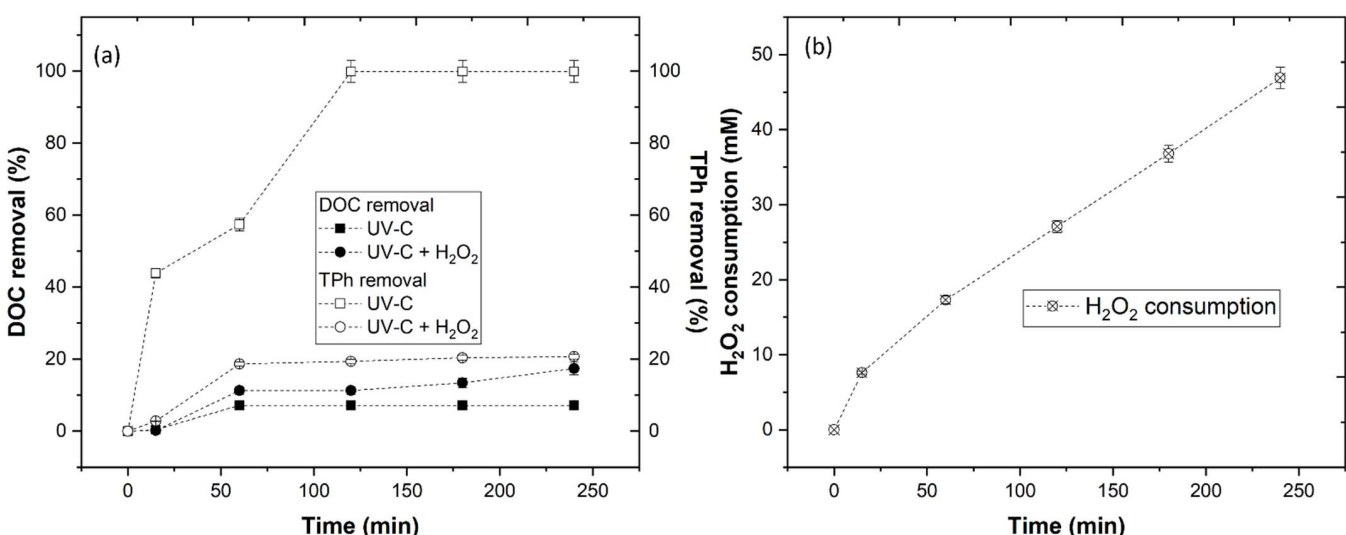

**Figure 12.** (**a**) Removal of DOC and TPh with the application of UV-C and UV-C + $H_2O_2$. (**b**) $H_2O_2$ consumption. AOPs operational conditions were as follows: pH = 3.0, radiation UV-C (254 nm), temperature = 298 K, agitation 350 rpm, time = 240 min.

### 3.3.4. Photo-Fenton-Based Processes

Considering the low DOC removal observed with Fenton, Fenton-like and heterogeneous Fenton, the addition of UV-C radiation could be a good strategy. In the work of Park et al. [59], the efficiency of the Fenton process was compared with photo-Fenton processes for the treatment of livestock wastewater, and the results showed significant differences between the two, with photo-Fenton presenting the highest removals. In the work of Alalm et al. [20], the results showed the high efficiency of the photo-Fenton process in the degradation of pesticides. Although working with different wastewaters, photo-Fenton processes showed high efficiency in the degradation of organic carbon.

In our study, Figure 13 shows a DOC reduction of 70.3, 51.1 and 54.2% for photo-Fenton, photo-Fenton-like and heterogeneous photo-Fenton processes, respectively. TPh removal was also evaluated, and the results showed the near complete removal of it after 240 min. The increase of DOC and TPh removal can be explained by three factors: (1) the hydrolysis of $H_2O_2$, leading to the production of $HO^\bullet$ radicals, (2) the reduction of $Fe^{3+}$ to $Fe^{2+}$ by UV radiation leads to the production of $HO^\bullet$ radicals and increases the concentration of $Fe^{2+}$ in solution (Equation (6)), which reacts more rapidly with $H_2O_2$ than $Fe^{3+}$, to yield more $HO^\bullet$ radicals than the Fenton-based processes without radiation, and (3) the speciation of $Fe^{3+}$ in the mono-aqueo complex occurs mainly at pH 3 [60].

$$Fe(HO)^{2+} + h\nu \rightarrow Fe^{2+} + HO^\bullet \tag{6}$$

The selection of the radiation type is also important. The UV-C lamp emits at a wavelength of 254 nm, which is ideal, considering that the photo-reduction of $Fe^{3+}$ to $Fe^{2+}$ is more efficient at a $\lambda < 600$ nm and the photolysis of $H_2O_2$ is more efficient at $\lambda = 254$ nm [61]. In the case of heterogeneous photo-Fenton, in comparison to homogeneous Fenton, the UV-C radiation appears to have potentiated the release of iron after 60 min of reaction. The slow release mechanism controlled the $Fe^{2+}$ concentration in the solution, decreasing the scavenger reactions observed in the photo-Fenton-like oxidation process (Equation (7)) [62], achieving higher DOC removal after 240 min.

$$Fe^{2+} + HO^\bullet \rightarrow Fe^{3+} + HO^- \tag{7}$$

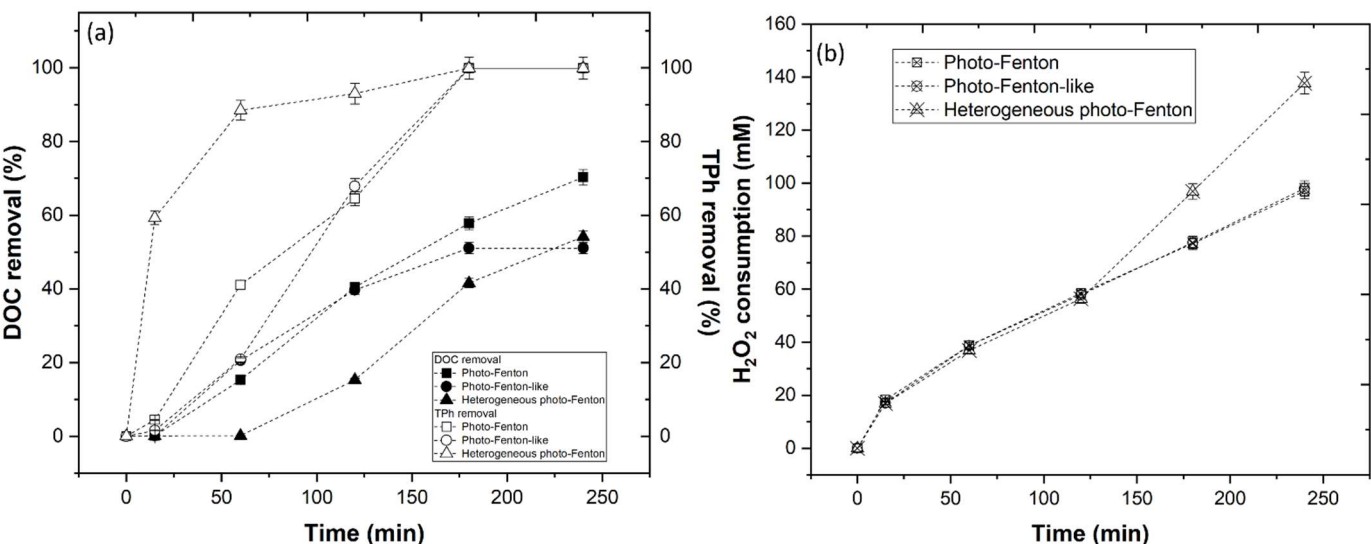

**Figure 13.** (**a**) Removal of DOC and TPh with the application of photo-Fenton, photo-Fenton-like and heterogeneous photo-Fenton. (**b**) $H_2O_2$ consumption. Photo-Fenton operational conditions were as follows: pH = 3.0, $[Fe^{2+}] = [Fe^{3+}]$ = 2.5 mM, radiation UV-C (254 nm), temperature = 298 K, agitation 350 rpm, time = 240 min.

### 3.3.5. Effect of CFD Combined with AOP

In the previous sections, it was observed that the application of photo-Fenton based processes increased DOC and TPh removal, but only to an extent. Also, considering the reduced efficiency of the CFD process in the removal of DOC and TPh, a complementary treatment is necessary; therefore, in this section, the pre-treated WW by CFD with *A. dealbata* process was submitted to three different photo-Fenton-based processes.

Figure 14 shows the evolution of DOC, TPh, and $H_2O_2$ consumption during the application of the photo-Fenton-based processes. As it can be observed, the initial application of LP led to a DOC and TPh reduction of 18.2 and 35.6%, respectively. With the application of photo-Fenton, photo-Fenton-like and heterogeneous photo-Fenton, a DOC removal of 98.2, 96.2 and 95.4%, respectively, was observed, with TPh removal reaching 99.9% after 240 min of reaction and $H_2O_2$ consumption reaching 117.7, 137.8 and 138.3 mM, respectively. The higher rate of degradation of total polyphenols in comparison to DOC by Fenton-based processes can be explained by the fact that when the $HO^\bullet$ attacks the phenol rings of hydroxybenzoic acids, hydroxycinnamic acids, flavonoids, and anthocyanins, the rings in these compounds break up to give organic acids and $CO_2$ [63,64].

These results show that the application of the LP as a pre-treatment reduced the turbidity, TSS, and TPh (Figure 15a), contributing to the removal of the dark color of the WW. The dark color and sediments block the UV light, preventing it from penetrating the water and decreasing its capacity to regenerate the $Fe^{3+}$ to $Fe^{2+}$, thereby decreasing the efficiency of the oxidation processes [65]. In addition, the removal of the suspended solids reduced the amount of organic carbon to be degraded by the photo-Fenton-based processes, which further increased the reaction's capacity allowing it to reach higher removal levels. The biodegradability ($BOD_5/COD$) was evaluated in this step, considering that this parameter determines the best approach for WW treatment (Figure 15b). As observed in Table 1, the WW presents a biodegradability index of 0.26, below 0; therefore, it is considered to be non-biodegradable [65]. The best approach to this treatment is the chemical methods applied in this work (CFD and Fenton-based-processes). With the application of CFD, Fenton, Fenton-like, and heterogeneous Fenton processes, the biodegradability index was observed to be lower than 0.3; thus, no biologic treatment process should be applied after. With the application of photo-Fenton, photo-Fenton-like, and heterogeneous photo-Fenton, the biodegradability index reached values >0.3; thus, a biologic treatment process could

be applied to decrease the remaining organic carbon left in the WW. The combination of CFD with AOPs had a biodegradability index <0.3; however, in this case, the near complete removal of organic carbon should be considered.

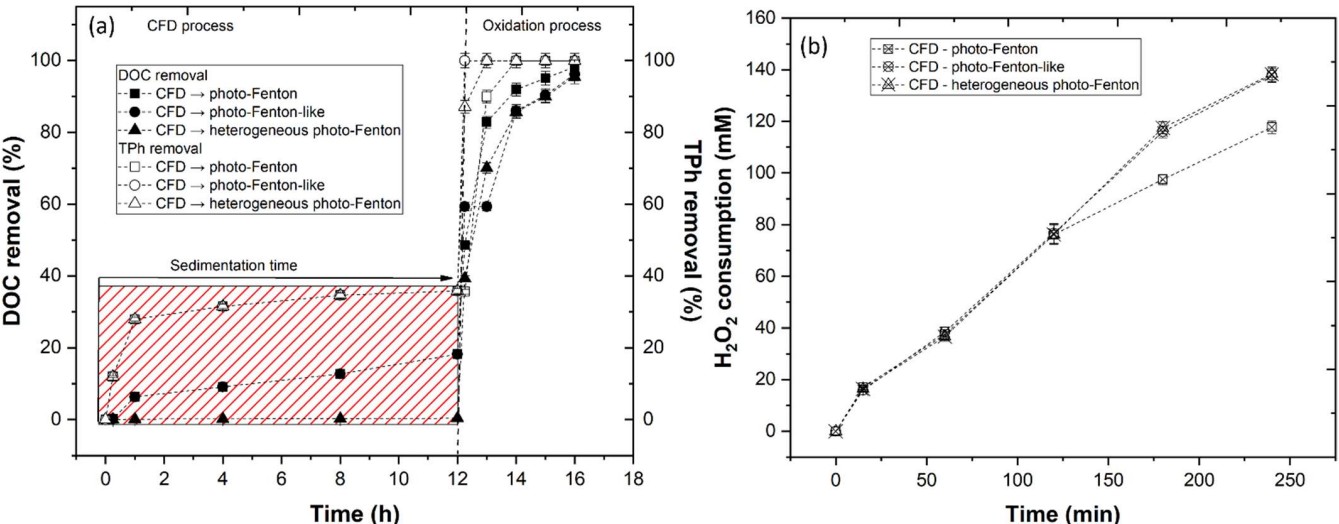

**Figure 14.** (**a**) Removal of DOC and TPh with the application of CFD-photo-Fenton, CFD-photo-Fenton-like and CFD-heterogeneous photo-Fenton. (**b**) $H_2O_2$ consumption.

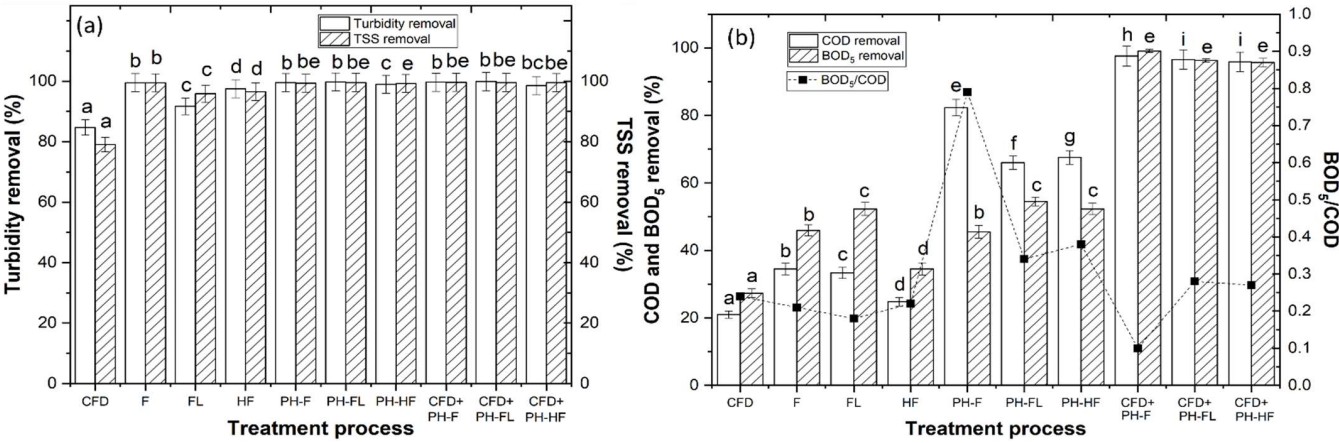

**Figure 15.** (**a**) Removal of turbidity and TSS, (**b**) removal of COD and $BOD_5$ and biodegradability ($BOD_5/COD$) from the application of different treatments. CFD—coagulation-flocculation-decantation, F—Fenton, FL—Fenton-like, HF—heterogeneous Fenton, PH-F—photo-Fenton, PH-FL—photo-Fenton-like, PH-HF—heterogeneous photo-Fenton. Means in bars with different letters represent significant differences ($p < 0.05$) within each parameter (turbidity, TSS, COD, and $BOD_5$) by comparing treatment processes.

To determine the final destiny of the treated wastewater, COD, $BOD_5$, and TSS was evaluated. In accordance with the Portuguese Law Decree no. 506/99, if the wastewater reaches values of COD $\leq$ 1000 mg $O_2$/L, $BOD_5$ $\leq$ 500 mg $O_2$/L, and TSS $\leq$ 700 mg/L, it can be disposed into a wastewater treatment plant (WWTP) as domestic wastewater. In accordance with the Law Decree no. 236/98, if the wastewater reaches values of COD $\leq$ 150 mg $O_2$/L, $BOD_5$ $\leq$ 40 mg $O_2$/L, and TSS $\leq$ 60 mg/L, the wastewater can be discharged into the environment. As observed in Figure 15, the application of the CFD, Fenton, Fenton-like and heterogeneous Fenton processes did not reach the legal values described in Law Decree no. 506/99; thus, are insufficient for WW treatment. The application of photo-Fenton, photo-Fenton-like, and heterogeneous photo-Fenton processes

reached the legal values described in Law Decree no. 506/99; thus, the subsequent treated water can be disposed into a WWTP as domestic wastewater. The application of combined CFD + photo-Fenton, CFD + photo-Fenton-like, and CFD + heterogeneous photo-Fenton reached the legal values in Law Decree no. 236/98, thus, this treated water can be used for irrigation purposes.

### 3.4. Kinetic Rate and Operational Costs Assessment

The different processes showed differences regarding their efficiency in the removal DOC from the WW. In order to better understand how DOC removal occurs as a function of time (t), the results were fitted into a pseudo first-order kinetic rate ($\ln[DOC]_t = -kt + \ln[DOC]_0$). Also, to gain better knowledge of the degradation of DOC, the time necessary to reduce it by 50% from the initial concentration of DOC–the half-life time ($t_{1/2} = 0.693/k$) is presented [66]. The results in Table 2 showed that Fenton, Fenton-like, and heterogeneous Fenton had the lowest kinetic rates. These processes, although having the capacity to produce $HO^\bullet$ radicals, showed a low capacity to regenerate the $Fe^{3+}$ to $Fe^{2+}$; thus, achieving lower kinetic rates. Also, it is necessary to consider the complex matrix of WW, which is a mixture of polyphenols, alcohols, organic acids, sugars, and other contaminants, which all increase the difficulty of treatment. Works, such as Benitez et al. [67], which have tested the degradation of single polyphenols such as gallic acid in water, observed high kinetic rates with the application of the Fenton process ($k = 0.132$ $min^{-1}$). From these results, the difficulty of treating real WW is realized.

**Table 2.** Pseudo first-order kinetic rate ($k$) and half-life ($t_{1/2}$) of DOC degradation.

| Processes | $k \times 10^{-3}$ ($min^{-1}$) | $t_{1/2}$ (min) | $R^2$ |
|---|---|---|---|
| Fenton | $1.31 \pm 5.3 \times 10^{-5}$ a | $529 \pm 5.9$ a | 0.920 |
| Fenton-like | $1.49 \pm 2.6 \times 10^{-5}$ b | $465 \pm 5.3$ b | 0.939 |
| Heterogeneous Fenton | $0.64 \pm 3.8 \times 10^{-6}$ c | $1081 \pm 5.9$ c | 0.920 |
| Photo-Fenton | $5.21 \pm 4.3 \times 10^{-5}$ d | $133 \pm 4.6$ d | 0.989 |
| Photo-Fenton-like | $3.38 \pm 2.6 \times 10^{-5}$ e | $205 \pm 5.6$ e | 0.941 |
| Heterogeneous photo-Fenton | $3.36 \pm 7.3 \times 10^{-5}$ e | $206 \pm 1.7$ e | 0.915 |
| CFD + photo-Fenton | $15.0 \pm 4.5 \times 10^{-5}$ f | $46 \pm 2.9$ f | 0.970 |
| CFD + photo-Fenton-like | $12.7 \pm 9.9 \times 10^{-5}$ g | $55 \pm 2.9$ g | 0.985 |
| CFD + heterogeneous photo-Fenton | $11.5 \pm 8.7 \times 10^{-5}$ h | $60 \pm 3.4$ g | 0.983 |

Means in the same column with different letters represent significant differences ($p < 0.05$) within each parameter by comparing the treatment processes.

With the direct application of photo-Fenton-based processes to real WW, the kinetic rate of DOC degradation increased significantly. The highest kinetic rate was achieved with the application of the photo-Fenton process. The photo-Fenton-like and heterogeneous photo-Fenton processes showed similar kinetic rates. On one hand, the application of $Fe^{3+}$ in the photo-Fenton-like process showed that the UV-C lamp has limits to the amount of $Fe^{3+}$ that is able to regenerate, so a part of the iron was precipitated as ferric hydroxide ($Fe(HO)_2^{2+}$), decreasing the reactions efficiency; however, the UV radiation clearly increased the regeneration of ferric iron, regarding the Fenton-like process. These results were in agreement with the work of Arslan-Alaton and Gurses [68], who observed that the application of UV radiation increased the kinetic rate of the photo-Fenton-like process, regarding Fenton-like in the degradation of procaine penicillin G. The increased kinetic rate observed between the heterogeneous Fenton and heterogeneous photo-Fenton processes, can be explained by the synergic effect between the photocatalysis of UV-C and ferrocene and the Fenton reaction between ferrous iron and $H_2O_2$. A similar result was obtained by Nie et al. [69] in the degradation of acid red B by heterogeneous photo-Fenton, with application of ferrocene.

From Table 2, it was observed that the application of pre-treatment CFD significantly increased the kinetic rate of the degradation of oxidation processes. The removal of

high percentages of suspended solids and turbidity had a synergic effect by removing particles that acted as radical scavengers. These results were in agreement with the work of Rodrigues et al. [7], who observed that pre-treatment of vinasse wastewater with CFD processes increased the efficiency of photo-Fenton process.

As previously observed, the application of the different reactors under the same operational conditions leads to different rates of DOC degradation. However, the efficiency of the reactor is not sufficient; it has to also be viable for application by companies treating agro-industrial wastewaters. Therefore, the energy efficiency (E) of the UV-C reactor was determined considering the total volume (Equation (8)).

$$Fe^{2+} + HO^{\bullet} \rightarrow Fe^{3+} + HO^{-} \tag{8}$$

where $DOC_0$ and $DOC_t$ represent DOC concentration at time 0 and t, P is the power of the UV-C lamp in kWh, t is the time in h, and V is the volume in $dm^{-3}$ [70]. To access the energy cost of an experiment, the inverse values of efficiency were multiplied by the value of electricity in Portugal (0.08 €/kWh). The reagent cost was also accessed, with reagents costs assumed as: 0.25 € $L^{-1}$ for $H_2O_2$ solution 33% (*w/v*), 0.72 € $kg^{-1}$ for $FeSO_4 \bullet 7H_2O$, 0.028 € $kg^{-1}$ for $FeCl_3 \bullet 6H_2O$, and 0.933 € $kg^{-1}$ for $C_5H_5)_2Fe$ [71].

In observation of Table 3, Fenton-based processes are the most economic, because no radiation is employed and, therefore, the costs are derived from the reagents employed. However, the kinetic rates are the lowest, and the final COD value does not allow the wastewater to be disposed as domestic wastewater. So, in this case, the cost of treatment does not reflect efficiency in terms of DOC degradation. In photo-Fenton based processes, the costs increased significantly due to the energy consumption of the lamp and the reagents cost. As observed in Table 3, the majority of the costs were provided from the application of the reagents, mainly from the application of $H_2O_2$. This consumption is a consequence of the synergic effect between: UV-C + $H_2O_2$ + catalyst. In this case, it is necessary to evaluate the UV absorption range of $H_2O_2$, $Fe^{2+}$, and $Fe^{3+}$, which is approximately 254 nm and is the same emission length of the UV-C mercury lamp [72]. Therefore, it is observed that a higher regeneration of $Fe^{3+}$ to $Fe^{2+}$ and one another means a higher conversion of $H_2O_2$ into $HO^{\bullet}$ radicals. The increase in operational cost is considered a necessary investment, as the final COD values allow for the disposal of the WW as domestic wastewater into a WWTP. Finally, the application of CFD had no real costs, considering the facility of growth and availability of the *A. dealbata* plant. The combination of processes allowed a significant decrease in energy costs. It was also observed a significant decrease of all costs regarding application of heterogeneous photo-Fenton. This cost is a necessary investment, as the final COD values allow for the disposal of the wastewater into the environment.

**Table 3.** Energy efficiency (E) and cost after 240 min of reaction, with P = 0.015 kW, t = 240 min. Means in the same column with different letters represent significant differences ($p < 0.05$) within each parameter by comparing the treatment processes. n.q.—not quantified.

| Processes | Energy Efficiency (E) (mg/L DOC/kWh) | Energy Cost (€/g/L DOC) | Reagent Cost (€/g/L DOC) | Total Cost (€/g/L DOC) |
|---|---|---|---|---|
| Fenton | n.q. | n.q. | $0.33 \pm 8.7 \times 10^{-5}$ a | $0.33 \pm 8.7 \times 10^{-5}$ a |
| Fenton-like | n.q. | n.q. | $0.08 \pm 1.0 \times 10^{-4}$ b | $0.08 \pm 1.0 \times 10^{-4}$ b |
| Heterogeneous Fenton | n.q. | n.q. | $0.34 \pm 1.4 \times 10^{-4}$ c | $0.34 \pm 1.5 \times 10^{-4}$ c |
| Photo-Fenton | $2201 \pm 5.7$ a | $3.63 \times 10^{-2} \pm 5.2 \times 10^{-5}$ a | $1.25 \pm 3.4 \times 10^{-4}$ d | $1.29 \pm 5.8 \times 10^{-4}$ d |
| Photo-Fenton-like | $1598 \pm 11.3$ b | $5.01 \times 10^{-2} \pm 3.7 \times 10^{-5}$ b | $1.26 \pm 1.8 \times 10^{-4}$ e | $1.31 \pm 5.4 \times 10^{-4}$ e |
| Heterogeneous photo-Fenton | $1692 \pm 14.2$ c | $4.73 \times 10^{-2} \pm 3.3 \times 10^{-5}$ c | $1.77 \pm 3.2 \times 10^{-4}$ f | $1.82 \pm 9.6 \times 10^{-5}$ f |
| CFD + photo-Fenton | $2515 \pm 9.4$ d | $3.18 \times 10^{-2} \pm 9.1 \times 10^{-5}$ d | $1.52 \pm 1.4 \times 10^{-3}$ g | $1.55 \pm 1.9 \times 10^{-4}$ g |
| CFD + photo-Fenton-like | $2452 \pm 6.7$ e | $3.26 \times 10^{-2} \pm 1.3 \times 10^{-4}$ e | $1.78 \pm 2.7 \times 10^{-4}$ f | $1.81 \pm 1.6 \times 10^{-4}$ h |
| CFD + heterogeneous photo-Fenton | $2428 \pm 8.1$ f | $3.29 \times 10^{-2} \pm 5.7 \times 10^{-5}$ f | $1.78 \pm 5.3 \times 10^{-4}$ h | $1.81 \pm 1.8 \times 10^{-4}$ i |

All the results commented on in this section point out the influence of the catalyst used, the reactor applied, and the type of wastewater used for the degradation of DOC. It became clear in this section that when comparisons between reactors are made, it is necessary to indicate exactly how they were performed. It was also found that it is different to evaluate the rate of degradation of DOC from raw WW than from coagulated WW. It was also necessary to show how the CFD process acted in the WW through the removal of turbidity, TSS, and VSS, which had a negative influence on the efficiency of the Fenton and photo-Fenton-based processes. Finally, it is necessary to point out the nature of the catalyst used and how it interacts with the radiation emitted by the UV lamp.

## 4. Conclusions

The initial proposal of this work was to conduct a series of processes focused not only on the treatment of WW but also on the recycling of the sludge and treated wastewater as fertilizer and irrigation water. For the separation of the sludge (solid part) from the wastewater (liquid part), it was concluded that the application of LP achieved a high volume of water recuperation and a low volume of sludge production. This sludge is concluded to be non-toxic, and it can be recycled as fertilizer. Several photo-Fenton-based processes were conducted on the raw WW, and it is concluded that the application of photo-Fenton, photo-Fenton-like, and heterogeneous photo-Fenton processes generate sufficient $HO^\bullet$ radicals to degrade TPh and COD to a level accepted by Portuguese legislation on the discharge of domestic wastewater to a WWTP, which is important for industries linked to a domestic grid. With the application of photo-Fenton-based processes to coagulated wastewater, the levels of TPh and COD reached those allowed by the recycling of water for irrigation purposes. This combined system is important for industries that are not linked to a domestic grid. Finally, a cost analysis revealed that the application of photo-Fenton-based processes directly to raw WW, or the application of photo-Fenton-based processes to coagulated wastewater, has low costs to industries, ranging between 1.29 and 1.82 €/g/L DOC. Regarding a global assessment, the results obtained in this work show that the application of LP is an ecologically-friendly and low-cost treatment process that allows for the recuperation of an invasive species (*A. dealbata*) and the recycling of the subsequent sludge at the same time. It is also concluded that the combination of CFD with photo-Fenton-based processes has economic advantages since most of the water consumed in the winery can be recycled for irrigation.

**Author Contributions:** Conceptualization, N.J. and A.R.T.; methodology, N.J.; software, N.J.; validation, N.J., A.R.T., M.S.L. and J.A.P.; formal analysis, N.J.; investigation, N.J.; resources, N.J.; data curation, N.J.; writing—original draft preparation, N.J.; writing—review and editing, N.J., M.S.L. and J.A.P.; visualization, N.J., M.S.L. and J.A.P.; supervision, J.A.P.; project administration, J.A.P.; funding acquisition, M.S.L. and J.A.P. All authors have read and agreed to the published version of the manuscript.

**Funding:** The authors are grateful for the financial support of the Project AgriFood XXI, operation no. NORTE-01-0145-FEDER-000041, and to the Fundação para a Ciência e a Tecnologia (FCT) for the financial support provided to CQVR through UIDB/00616/2020. Ana R. Teixeira also thanks the FCT for the financial support provided through the doctoral scholarship UI/BD/150847/2020.

**Data Availability Statement:** The data presented in this study are available on request from the corresponding author.

**Conflicts of Interest:** The authors declare no conflict of interest.

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
