# Peer review of "Agro-Industrial Wastewater Treatment with Acacia dealbata Coagulation/Flocculation and Photo-Fenton-Based Processes"

_recycling, doi:10.3390/recycling7040054_

Round 1

Reviewer 1 Report

The paper entitled "Agro-industrial wastewater treatment with Acacia dealbata coagulation/flocculation and photo-Fenton based processes" describes the adoption of the biomass of an invasive plant for the treatment of winery wastewaters (WW). To complete the WW treatment the Authors also tested Fe-based and UV-based AOPs. The paper is very interesting and well written. Consequently, I suggest the following minor revisions:

- Why the Authors adopted UVC lamps instead of UVA lamps during the Photo-Fenton based experiments? It also works with UVA radiation. Consequently, the Authors could have used solar radiation too! Please, support your choice in the experimental section.

- from line 249: please correct the numeration of the sections!

- section 3.2.2: the Authors should carry out a further experiment to support the discussion. It is very simple: 1) suspend 400 mg/L LP in distilled water at pH=3; 2) monitor the DOC concentration released by LP in water at 1, 4, 8, and 12 h.

- please, adjust the scale of the y axis of the figures (6a-b, 7a and 10a) with respect to the values found! 

- line 407: low efficiencies?

- line 452 / reaction 6: please, report in the manuscript that the speciation of Fe(III) in the mono-aqueo complex occurs mainly at pH 3. It is important!

Author Response

The changes performed for Reviewer 1 are highlighted in the article in yellow, Reviewer 2 are highlighted in green, and Reviewer 3 are highlighted in blue.

Comments and Suggestions for Authors

The paper entitled "Agro-industrial wastewater treatment with Acacia dealbata coagulation/flocculation and photo-Fenton based processes" describes the adoption of the biomass of an invasive plant for the treatment of winery wastewaters (WW). To complete the WW treatment the Authors also tested Fe-based and UV-based AOPs. The paper is very interesting and well written. Consequently, I suggest the following minor revisions:

1) Why the Authors adopted UVC lamps instead of UVA lamps during the Photo-Fenton based experiments? It also works with UVA radiation. Consequently, the Authors could have used solar radiation too! Please, support your choice in the experimental section.

Dear reviewer, although some works shows the potential application of UV-A and solar radiation as more friendly technologies for wastewater treatment, in this work the main purpose was assess the use of an invasive specie as a coagulant. As a complementary treatment were used Fenton-based processes with UV-C radiation. However, we agree with the reviewer that UV-A and/or solar radiation could be also studied with similar results. This justification was added to the article in line 443 – 445.

2) From line 249: please correct the numeration of the sections!

The numeration of the sections was corrected.

3) section 3.2.2: the Authors should carry out a further experiment to support the discussion. It is very simple: 1) suspend 400 mg/L LP in distilled water at pH=3; 2) monitor the DOC concentration released by LP in water at 1, 4, 8, and 12 h.

Thank you for the observations. Indeed, the application of different concentrations of LP in water helps to support the discussion. In line 309 – 317 it is explained the results of the LP mechanism in water. These results are also shown in Figure 7(e) and Figure 7(f).

4) please, adjust the scale of the y axis of the figures (6a-b, 7a and 10a) with respect to the values found!

The y axis of the figures was changed.

5) line 407: low efficiencies?

The authors wanted to say that the efficiency of heterogeneous Fenton in the removal of DOC and TPh was lower than Fenton and Fenton-like processes. This sentence was changed in line 429 – 430.

6) line 452 / reaction 6: please, report in the manuscript that the speciation of Fe(III) in the mono-aqueo complex occurs mainly at pH 3. It is important!

The justification was added to the article in line 477 – 478.

Reviewer 2 Report

Article entitled Agro-industrial wastewater treatment with Acacia dealbata coagulation/flocculation and photo-Fenton based processes written by Nuno Jorge, Ana R. Teixeira, Marco S. Lucas and José A. Peres and submitted to Recycling journal as a draft no 1835325 deals with an important issue of wastewater treatment.

The article is interesting and could be considered for publication in Recycling journal. As English is not my native language, I am not able to assess language correctness. However, while reading I found some statements missing, confusing or unclear. Below I enclose the list of my comments.

Literature review is incomplete. Authors discussing the topics of wastewater treatment, coagulants etc using 18 literature items only. The current state of knowledge should be accurately presented.

Hydrogen peroxide is well known COD determination disruptor. How the Authors remove hydrogen peroxide on COD?

I suggest moving some text from Materials and Methods to Results chapter. In M&M should be procedure only. Lines 160-168, Fig. 3 are clear results. Similarly, Fig. 4.

How Fenton (and Fenton-like) process was terminated? Coagulation of iron in pH 7 may not be 100% in case of Fe removal and residual hydrogen peroxide (strong oxidant) still will be present in solution, causing decomposition of organics.

There is problem with chapters numbering 2.3 is after 2.6. Results and 3.1 are missing.

Some repetitions are present in the text eg lines 357 – 365.

Why the proposed doses of regents were used? Why the Authors do not find optimal reagent doses? Why other pH were not tested? What was used as heterogenaous Fenton catalyst? All of this should be clarified in M&M.

The article was prepared quite sloppy, it should first be corrected in terms of editing. For this reason, reading it and checking it in terms of its content is very difficult. Based on my comments and general impression I suggest major revision.

Author Response

The changes performed for Reviewer 1 are highlighted in the article in yellow, Reviewer 2 are highlighted in green, and Reviewer 3 are highlighted in blue.

Comments and Suggestions for Authors

Article entitled Agro-industrial wastewater treatment with Acacia dealbata coagulation/flocculation and photo-Fenton based processes written by Nuno Jorge, Ana R. Teixeira, Marco S. Lucas and José A. Peres and submitted to Recycling journal as a draft no 1835325 deals with an important issue of wastewater treatment.

The article is interesting and could be considered for publication in Recycling journal. As English is not my native language, I am not able to assess language correctness. However, while reading I found some statements missing, confusing or unclear. Below I enclose the list of my comments.

1) Literature review is incomplete. Authors discussing the topics of wastewater treatment, coagulants etc using 18 literature items only. The current state of knowledge should be accurately presented.

Dear Reviewer, thank you for your comments. Clearly the state of art needed improvement. The changes were performed in the introduction, pointing out the composition of the winery wastewater matrix. It was also added information regarding the process of photo-Fenton. It was also pointed out the difficulties of treating these types of matrixes.

2) Hydrogen peroxide is well known COD determination disruptor. How the Authors remove hydrogen peroxide on COD?

The reactions were quenched with the application of sodium sulfite anhydrous, thus the H2O2 did not influenced the COD analysis (line 195). It is also pointed out that DOC analysis was performed because, DOC is not influenced by residual H2O2 that can remain in the solution.

3) I suggest moving some text from Materials and Methods to Results chapter. In M&M should be procedure only. Lines 160-168, Fig. 3 are clear results. Similarly, Fig. 4.

Thank you for the suggestion. The results referent to the characterization of leaves powder were changed to the results and discussion section. They are found in line 235 – 262 (highlighted in green).

4) How Fenton (and Fenton-like) process was terminated? Coagulation of iron in pH 7 may not be 100% in case of Fe removal and residual hydrogen peroxide (strong oxidant) still will be present in solution, causing decomposition of organics.

As mentioned earlier, the Fenton and Fenton-like experiments were terminated with the application of sodium sulfite anhydrous, which removed the H2O2 from solution, thus ending the production of hydroxyl radicals and consequently ending the degradation process. An analysis by AAS performed after increasing the pH to 7.0 showed a complete removal of iron from solution.

5) There is problem with chapters numbering 2.3 is after 2.6. Results and 3.1 are missing.

Thank you for pointing out these problems. The sections numeration was corrected.

6) Some repetitions are present in the text eg lines 357 – 365.

Thank you for your observations. The sentence in line 383 – 387 was changed to avoid repetitions.

7) Why the proposed doses of regents were used? Why the Authors do not find optimal reagent doses? Why other pH were not tested? What was used as heterogeneous Fenton catalyst? All of this should be clarified in M&M.

Dear reviewer, the doses of reagents were optimized in previous works. The pH was varied from 3 to 9 and the Fe2+ concentration was varied from 0.5 to 10 mM. The results showed that pH 3.0 is the most efficient pH. Results also showed that 2.5 mM is the best Fe2+ concentration, that > 2.5 mM, scavenging reaction occurs, and 2.5 mM < the reactions are slower, with lower kinetic rate of DOC degradation.

As heterogeneous catalyst, it was used ferrocene, because, unlike ferrous sulfate or ferric chloride, ferrocene does not dissolve in the wastewater, been able to be recovered. This question is answered in line 190 – 194.

Reviewer 3 Report

Line 54. I think It is not adequate to point out that Acacia based coagulants have never been used before to treat wastewater. It should be at least few references suggesting the possibility of using Acacia leaves powder as coagulant.

Line 84-85. I thin some information is missing in the sentence.

There is information missing between section 2.7 and 3.2.  Section 3.1 is missing and title of section 3 does not appear.

Author Response

The changes performed for Reviewer 1 are highlighted in the article in yellow, Reviewer 2 are highlighted in green, and Reviewer 3 are highlighted in blue.

Comments and Suggestions for Authors

 1) Line 54. I think It is not adequate to point out that Acacia based coagulants have never been used before to treat wastewater. It should be at least few references suggesting the possibility of using Acacia leaves powder as coagulant.

Dear Reviewer, thank you for your observations. New works were added, mentioning the application of Acacia based coagulants as products for wastewater treatment. It is also mentioned that based in these works, it is possible to apply the A. dealbata leaves as a coagulant, in line 55 – 61.

2) Line 84-85. I think some information is missing in the sentence.

The aim and novelty of the present work was changed, and the major points of this work were highlighted in line 94 – 99.

3) There is information missing between section 2.7 and 3.2.  Section 3.1 is missing and title of section 3 does not appear.

Dear Reviewer, thank you for your observations. The sections titles were corrected.

Round 2

Reviewer 2 Report

This is my second review of this article. The Authors answered all of my comments. Suggested corrections have been applied. Second version of the manuscript is much better than the first one. I suggest to accept this article in its present form.

Author Response

Comments and Suggestions for Authors

This is my second review of this article. The Authors answered all of my comments. Suggested corrections have been applied. Second version of the manuscript is much better than the first one. I suggest to accept this article in its present form.

The comments and suggestions of Reviewer 2 were essential to improve the quality of this article. The authors are grateful for the help provided by Reviewer 2.

Reviewer 3 Report

Line 70. The meaning of AOP abbreviation should be indicated.

A general explanation of the different tests and results to be presented at the beginning of section3 could facilitate readers comprehension.

There are some typos in the manuscript.

Author Response

Reviewer 3 – Round 2

Comments and Suggestions for Authors

The changes performed in the manuscript were highlighted in yellow.

(1) Line 70. The meaning of AOP abbreviation should be indicated.

Thank you for this observation, the authors indicated the meaning of AOPs in line 70.

(2) A general explanation of the different tests and results to be presented at the beginning of section3 could facilitate readers comprehension.

The authors provided a general explanation of this work in line 236 – 247, at the beginning of section 3, to facilitate readers comprehension.

(3) There are some typos in the manuscript.

Thank you reviewer 3 for your observations, some typos were corrected in the manuscript.